# Structural and mechanistic insights into ribosomal ITS2 RNA processing by nuclease-kinase machinery

Jiyun Chen[1†], Hong Chen[1†], Shanshan Li[2†], Xiaofeng Lin[1], Rong Hu[1], Kaiming Zhang[2]*, Liang Liu[1]*

[1]State Key Laboratory of Cellular Stress Biology, School of Life Sciences, Faculty of Medicine and Life Sciences, Xiamen University, Xiamen, China; [2]MOE Key Laboratory for Cellular Dynamics and Division of Life Sciences and Medicine, University of Science and Technology of China, Hefei, China

*For correspondence:
kmzhang@ustc.edu.cn (KZ);
liangliu2019@xmu.edu.cn (LL)

[†]These authors contributed equally to this work

**Abstract** Precursor ribosomal RNA (pre-rRNA) processing is a key step in ribosome biosynthesis and involves numerous RNases. A HEPN (higher eukaryote and prokaryote nucleotide binding) nuclease Las1 and a polynucleotide kinase Grc3 assemble into a tetramerase responsible for rRNA maturation. Here, we report the structures of full-length *Saccharomyces cerevisiae* and *Cyberlindnera jadinii* Las1-Grc3 complexes, and *C. jadinii* Las1. The Las1-Grc3 structures show that the central coiled-coil domain of Las1 facilitates pre-rRNA binding and cleavage, while the Grc3 C-terminal loop motif directly binds to the HEPN active center of Las1 and regulates pre-rRNA cleavage. Structural comparison between Las1 and Las1-Grc3 complex exhibits that Grc3 binding induces conformational rearrangements of catalytic residues associated with HEPN nuclease activation. Biochemical assays identify that Las1 processes pre-rRNA at the two specific sites (C2 and C2'), which greatly facilitates rRNA maturation. Our structures and specific pre-rRNA cleavage findings provide crucial insights into the mechanism and pathway of pre-rRNA processing in ribosome biosynthesis.

## eLife assessment

This study represents a **valuable** mechanistic contribution towards understanding how ribosomal RNA is processed during ribosome biogenesis. The biochemical evidence supporting the major conclusions is **convincing**. This work will be of interest to cell biologists and biochemists working on ribosome biogenesis.

## Introduction

Ribosomes are large molecular machines assembled from numerous proteins and RNAs that are responsible for protein synthesis in cells (*Anger et al., 2013*; *Gasse et al., 2015*; *Khatter et al., 2015*). In eukaryotes, ribosome biosynthesis is tightly coupled to cell growth and cell cycle progression and is critical for regulating normal cell size and maintaining cell cycle progression (*Castle et al., 2013*). Ribosome biosynthesis is an extremely complicated process involving about 200 assembly and processing factors, which are involved in a series of continuous assembly and processing reactions such as ribosome protein folding, modification, assembly, and precursor rRNA (pre-rRNA) processing (*Gasse et al., 2015*; *Lafontaine, 2015*; *Pillon et al., 2017*; *Wu et al., 2016*). Mature ribosomes in yeast *Saccharomyces cerevisiae* contain 79 proteins and four RNAs (25S, 18S, 5.8S, and 5S rRNA) (*Doudna and Rath, 2002*; *Wilson and Doudna Cate, 2012*; *Woolford and Baserga, 2013*). 5S rRNA is transcribed by RNA polymerase III, whereas 25S, 18S, and 5.8S rRNA are cotranscribed by RNA

polymerase I as a single long precursor (35S pre-rRNA) (*Tomecki et al., 2017*). Except 25S, 18S, and 5.8S rRNA sequences, the 35S pre-rRNA also includes 5'-external transcribed spacer sequence (ETS), 3'-external transcribed spacer sequence, and two internal transcribed spacer sequences (ITS1 and ITS2) (*Fromm et al., 2017*). ITS1 is located between 5.8S and 18S rRNA, and ITS2 is located between 5.8S and 25S rRNA (*Coleman, 2003*; *Côté et al., 2002*). The mature rRNAs are generated by a large number of endonucleases and exonucleases that remove these transcribed spacers step by step through multiple efficient and correct processing reactions (*Granneman et al., 2011*). The pre-rRNA processing factors synergistically produce mature rRNAs and lead to an accurate and efficient assembly of mature ribosome in the nucleolus, which are key to cell survival (*Pillon and Stanley, 2018*). Mutations in the genes that encode these pre-rRNA processing factors are often lethal (*Tomecki et al., 2017*). Although numerous evolutionarily conserved protein factors have been found to be involved in the processing and modification of ribosomal RNA, the detailed pathways of these factors and their specific processing mechanisms are not well understood.

Las1 and Grc3 are highly conserved proteins and have recently been identified as core enzymes involved in processing and removing ITS2 spacer, which is a key step in the synthesis of 60S ribosomal subunit (*Gasse et al., 2015*; *Schillewaert et al., 2012*). Las1 is characterized as a nucleolar protein essential for ribosome biogenesis, as well as cell proliferation and cell viability in *S. cerevisiae* (*Castle et al., 2012*; *Castle et al., 2010*; *Doseff and Arndt, 1995*). It is important to determine the role of Las1 in rRNA metabolic pathways and regulatory networks associated with ribosome biogenesis and cell proliferation. Recent studies have demonstrated that Las1 is an endoribonuclease that contains a HEPN (higher eukaryote and prokaryote nucleotide binding) domain responsible for rRNA processing (*Pillon et al., 2020*; *Pillon et al., 2017*). The HEPN domain must be dimerized to form an active nuclease, such as the Cas13 effectors in CRISPR immune defense systems, whose catalytic site is formed by two HEPN domains involved in non-specific cleavage of single-stranded RNA (*Knott et al., 2017*; *Liu et al., 2017a*; *Liu et al., 2017b*; *Zhang et al., 2018*). Interestingly, Las1 specifically cleaves at the C2 site within ITS2 and generates the 7S pre-rRNA and 26S pre-rRNA (*Fernández-Pevida et al., 2015*). It is not clear why Las1 HEPN nuclease specifically targets and cleaves ITS2 only at the C2 site, and whether this cleavage depends on a specific sequence or secondary structure. In addition, the cleavage activity of Las1 is primarily dependent on another enzyme Grc3. Grc3 is a polynucleotide kinase responsible not only for Las1 nuclease activation, but also for nonspecific phosphorylation of the 5'-OH of the 26S pre-rRNA produced by Las1 cleavage, providing a signal for further processing by Rat1-Rai1 exonuclease (*Gordon et al., 2019*; *Xiang et al., 2009*). Although cryo-electron microscopy (cryo-EM) reveals cross-linked *Chaetomium thermophilum* (Ct) Las1 and Grc3 assemble into a super-dimer, due to flexibility, critical structural information is missing for the coiled-coil (CC) domain of Las1 and the N-terminal and C-terminal regions of Grc3 (*Pillon et al., 2019*). It remains unknown how Las1 and Grc3 coordinate with each other in terms of substrate binding and nuclease activation.

In this study, we identified that *S. cerevisiae* (Sc) Las1 endoribonuclease initially cleaves ITS2 in a step-by-step fashion at two specific sites, which greatly promotes the maturation of 25S rRNA. Additionally, we solved the crystal structures of full-length ScLas1-Grc3 complex and *Cyberlindnera jadinii* (Cj) Las1-Grc3 complex, as well as the high-resolution structure of CjLas1 HEPN domain. Our structural and biochemical findings uncovered a detailed mechanism of polynucleotide kinase-mediated activation of HEPN nuclease, providing a molecular basis for clearly understanding the process of pre-rRNA processing and maturation in ribosome biosynthesis.

## Results

### Las1 cleaves ITS2 at two specific sites

HEPN endoribonucleases such as Cas13, Ire1, and RNase L are metal-independent RNA-specific enzymes that efficiently cleave substrate RNA at multiple sites (*Abudayyeh et al., 2016*; *Huang et al., 2014*; *Korennykh et al., 2009*; *Lee et al., 2008*; *Wang et al., 2014*). Las1 is also identified as a HEPN-containing RNase, but it has only been found to initially cut ITS2 RNA at a single specific position (C2). To investigate whether there are other potential sites for Las1 cleavage in ITS2, we performed in vitro RNA cleavage assays using a 5'-Cy5- and 3'-Cy3-labeled 33-nt ITS2 RNA substrate (*Figure 1A*). Las1 shows weak or no detectable activity to ITS2 in the absence of Grc3, but exhibits robust activity in the presence of Grc3 (*Figure 1B*, *Figure 1—figure supplement 1*), revealing Grc3-dependent Las1

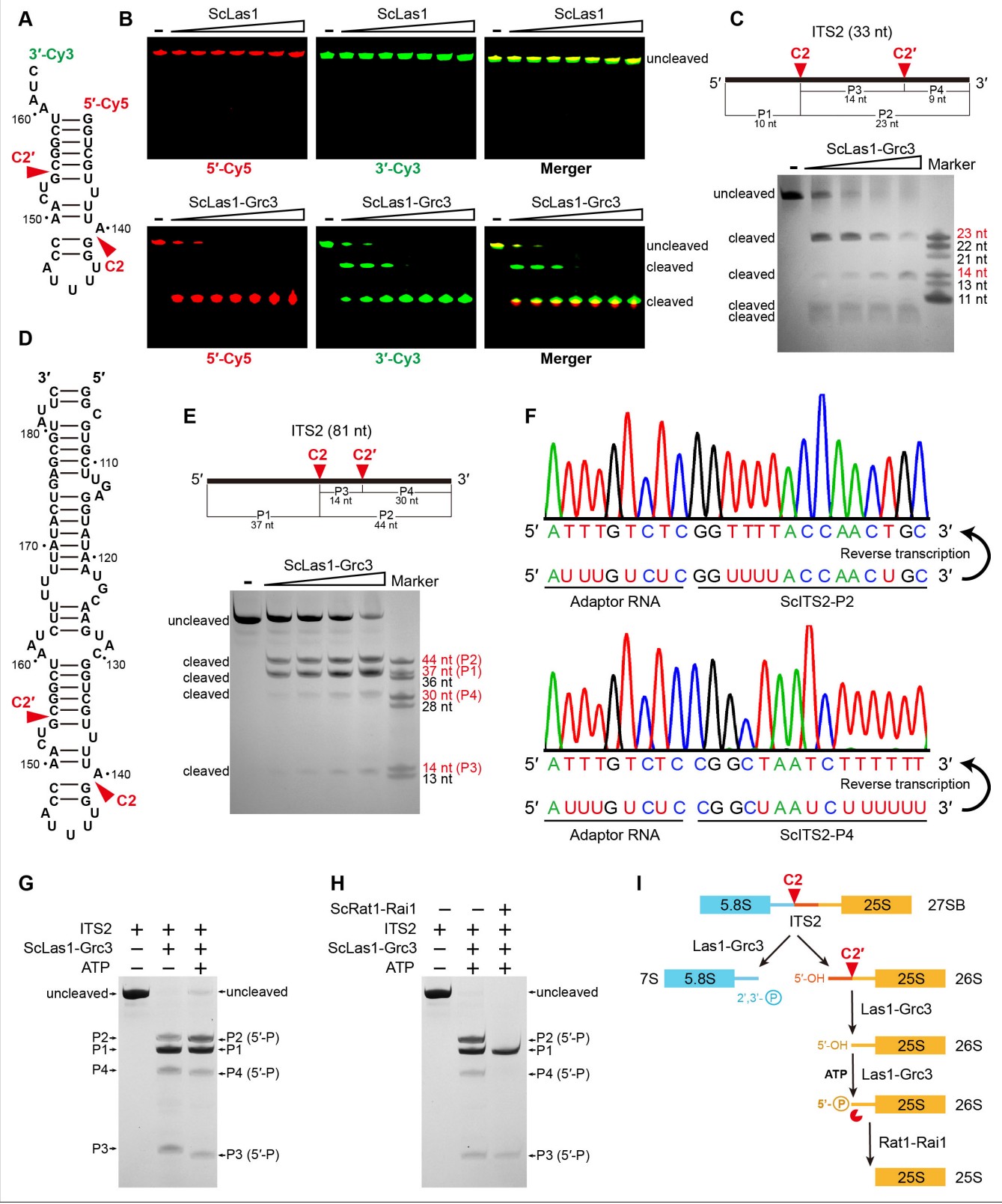

**Figure 1.** ScLas1 specifically cleaves ITS2 at the C2 and C2' sites. (**A**) 33-nt ITS2 RNA with 5'-Cy5 and 3'-Cy3 labels. (**B**) In vitro RNA cleavage assay using 5'-Cy5 and 3'-Cy3-labeled 33-nt RNA. (**C**) In vitro RNA cleavage assay of unlabeled 33-nt RNA. (**D**) 81-nt ITS2 RNA. (**E**) In vitro RNA cleavage assay of unlabeled 81-nt RNA. (**F**) RNA-sequencing traces from ScLas1-cleaved ITS2 products P2 and P4. (**G**) RNA phosphorylation assay with ScLas1-Grc3

*Figure 1 continued on next page*

*Figure 1 continued*

complex. (**H**) RNA degradation assay with ScRat1-Rai1 complex. (**I**) The ITS2 pre-rRNA processing pathway. All cleavage experiments were repeated three times.

The online version of this article includes the following source data and figure supplement(s) for figure 1:

**Source data 1.** Original files for the RNA cleavage analysis in *Figure 1B* (5′-Cy5, 3′-Cy3, Merger).

**Source data 2.** Original scans of the relevant RNA cleavage analysis in *Figure 1B* (5′-Cy5, 3′-Cy3, Merger) with band and sample labels.

**Source data 3.** Original file for the RNA cleavage analysis in *Figure 1C*.

**Source data 4.** Original scan of the relevant RNA cleavage analysis in *Figure 1C* with band and sample labels.

**Source data 5.** Original file for the RNA cleavage analysis in *Figure 1E*.

**Source data 6.** Original scan of the relevant RNA cleavage analysis in *Figure 1E* with band and sample labels.

**Source data 7.** Original file for the RNA cleavage analysis in *Figure 1G and H*.

**Source data 8.** Original scan of the relevant RNA cleavage analysis in *Figure 1G and H* with band and sample labels.

**Figure supplement 1.** CjGrc3-activated CjLas1-catalytic ITS2 pre-rRNA cleavage.

**Figure supplement 1—source data 1.** Original files for the RNA cleavage analysis in *Figure 1—figure supplement 1* (5′-Cy5, 3′-Cy3, Merger).

**Figure supplement 1—source data 2.** Original scans of the relevant RNA cleavage analysis in *Figure 1—figure supplement 1* (5′-Cy5, 3′-Cy3, Merger) with band and sample labels.

**Figure supplement 2.** ScGrc3 has no ITS2 pre-rRNA cleavage activity.

**Figure supplement 2—source data 1.** Original files for the RNA cleavage analysis in *Figure 1—figure supplement 2* (5′-Cy5, 3′-Cy3, Merger).

**Figure supplement 2—source data 2.** Original scans of the relevant RNA cleavage analysis in *Figure 1—figure supplement 2* (5′-Cy5, 3′-Cy3, Merger) with band and sample labels.

**Figure supplement 3.** Catalytic residues of Las1 HEPN domain are necessary for ITS2 pre-rRNA cleavage.

**Figure supplement 3—source data 1.** Original files for the RNA cleavage analysis in *Figure 1—figure supplement 3* (5′-Cy5, 3′-Cy3, Merger).

**Figure supplement 3—source data 2.** Original scans of the relevant RNA cleavage analysis in *Figure 1—figure supplement 3* (5′-Cy5, 3′-Cy3, Merger) with band and sample labels.

**Figure supplement 4.** Characterization of the metal independence of ITS2 pre-rRNA cleavage.

**Figure supplement 4—source data 1.** Original file for the metal-independent RNA cleavage analysis in *Figure 1—figure supplement 4*.

**Figure supplement 4—source data 2.** Original scan of the relevant metal-independent RNA cleavage analysis in *Figure 1—figure supplement 4* with band and sample labels.

nuclease activation. Interestingly, we observed a single 5′-Cy5-labeled cleavage product, and two prominent 3′-Cy3-labeled cleavage products including a final product and an intermediate product (*Figure 1B*), suggesting that cleavage of ITS2 substrate occurs at two specific sites. The 33-nt ITS2 substrate RNA harbors the C2 site, which is located between nucleotides A140 and G141 (*Figure 1A*). In addition to the C2 site, there is another specific position in ITS2 that is able to be processed by Las1. ScLas1 cleaves the 33-nt ITS2 at the C2 site to theoretically generate a 10-nt 5′-terminal product and a 23-nt 3′-terminal product (*Figure 1A*). Our merger data shows that the final 5′-terminal and 3′-terminal product bands are at nearly the same horizontal position on the gel (*Figure 1B*), indicating that they are very close in length. Therefore, we hypothesize that the 23-nt 3′-terminal product is an intermediate that can be further processed by Las1 at a specific site to produce two small products. We then mapped the cleavage products using the 33-nt ITS2 RNA without 5′-Cy5- and 3′-Cy3-label. We observed four cleavage bands, two approximately 23-nt (P2) and 14-nt (P3) in length, and two (P1 and P4) less than 11-nt in length (*Figure 1C*). The 23-nt product (P2) is obviously an intermediate cleavage product, which is further cleaved to generate 14-nt and 9-nt products. Based on these observations, we identify that another cleavage site, which we designated as C2′, is located between nucleotides G154 and C155. Las1 is able to process ITS2 at the C2 and C2′ sites in a step-by-step manner, resulting in a 9-nt 3′-end product and a 10-nt 5′-end product.

We obtained similar cleavage results with a longer 81-nt ITS2 RNA substrate (*Figure 1D and E*). To further confirm the cleavage site of C2′, we then mapped the cleavage sites of the 81-nt ITS2 using reverse transcription coupling sequencing methods (*Figure 1F*). We used an adaptor RNA to separately link the product RNA fragments to form template RNAs. After reverse transcription and sequencing of these template RNAs, we obtained the accurate sequence information for products P2

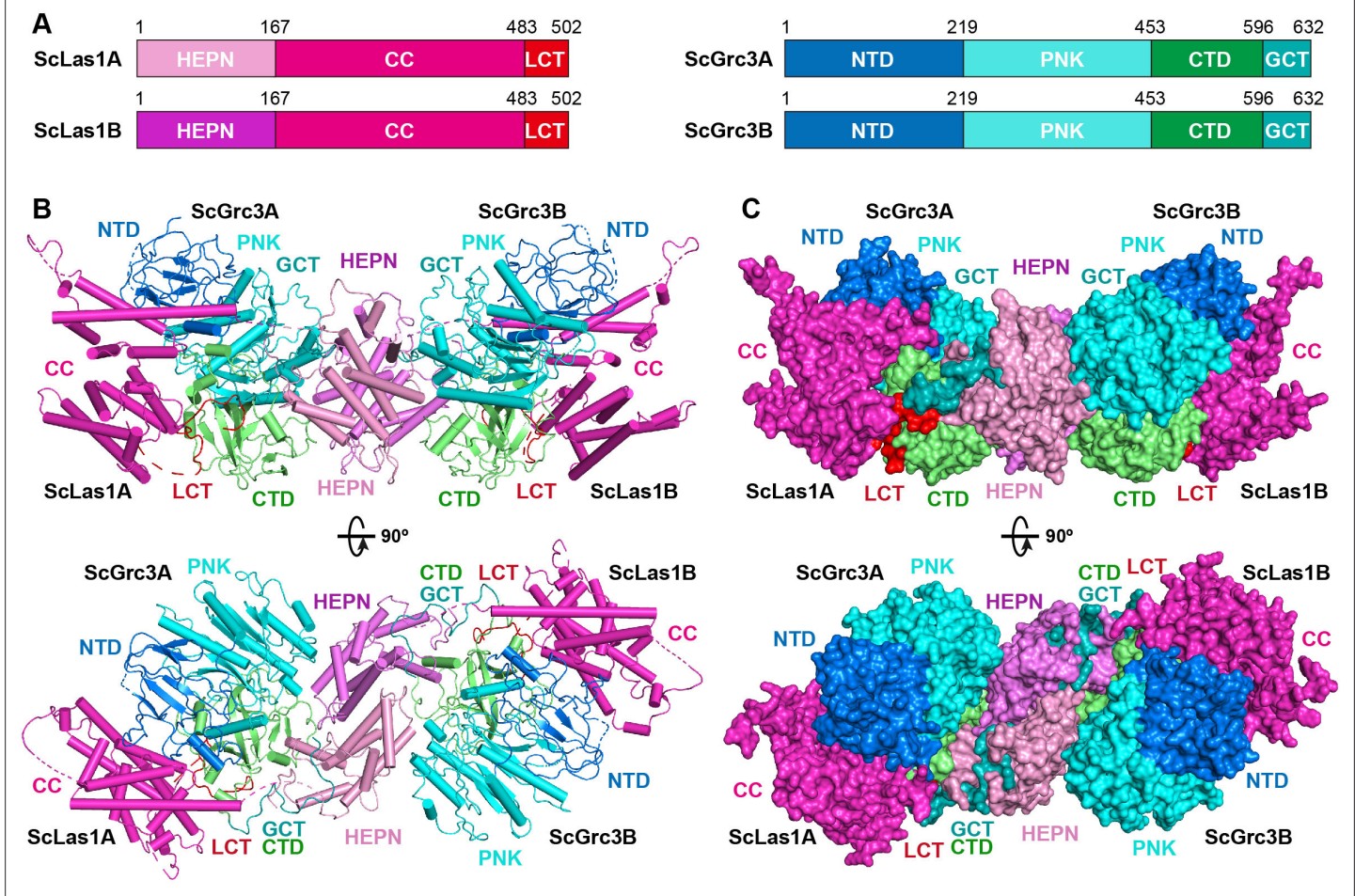

**Figure 2.** Overall structure of ScLas1-Grc3 complex. (**A**) Domain organization of ScLas1 and ScGrc3. (**B**) Ribbon representations of ScLas1-Grc3 complex. Color coding used for Las1 and Grc3 is identical to that used in (**A**). (**C**) Surface representations of ScLas1-Grc3 complex. Color coding used for Las1 and Grc3 is identical to that used in (**A**).

The online version of this article includes the following figure supplement(s) for figure 2:

**Figure supplement 1.** Single-particle cryo-electron microscopy (cryo-EM) analysis of the ScLas1-Grc3 complex.

and P4, revealing that the C2 site is located between nucleotides A140 and G141, and the C2′ site is located between nucleotides G154 and C155.

The cleavage products P2, P3, and P4 are all phosphorylated by Grc3 as they all show a slight shift when in the presence of ATP in the reaction (*Figure 1G*). This indicates that Las1 generates 5′-OH terminus following both C2 and C2′ cleavage. The phosphorylated products are further degraded by Rat1-Rai1 exonuclease, especially P2 and P4, which are completely degraded under given experimental conditions (*Figure 1H*). P3 is likely too short to degrade completely.

In addition, Grc3 shows no cleavage activity to ITS2 (*Figure 1—figure supplement 2*), and mutations in the HEPN catalytic residues of Las1 abolish C2 and C2′ cleavage (*Figure 1—figure supplement 3*), further confirming that both C2 and C2′ cleavages are attributed to metal-independent Las1 RNase (*Figure 1—figure supplement 4*).

## Overall structures of ScLas1-Grc3 complex and CjLas1-Grc3 complex

To elucidate how Las1 and Grc3 cooperate to direct ITS2 cleavage and phosphorylation, we determined the cryo-EM and crystal structures of full-length ScLas1-Grc3 complex at 3.07 Å and 3.69 Å resolution (*Figure 2, Figure 2—figure supplement 1, Tables 1 and 2*), respectively. Cleavage assay indicates that ScGrc3 can activate ScLas1 nuclease well for ITS2 cleavage, while CjGrc3 exhibits relatively weak activation ability for CjLas1 nuclease (*Figure 1B, Figure 1—figure supplement 1*). To

**Table 1.** Crystallographic data collection and refinement statistics.

| | ScLas1-Grc3 | CjLas1-Grc3 | CjLas1 |
|---|---|---|---|
| Data collection* | | | |
| Space group | C2 | C222$_1$ | P2$_1$2$_1$2$_1$ |
| *Cell dimensions* | | | |
| a, b, c (Å) | 233.6, 116.1, 159.3 | 152.6, 240.0, 237.0 | 51.5, 59.0, 158.7 |
| α,β,γ (°) | 90.0, 96.4, 90.0 | 90.0, 90.0, 90.0 | 90.0, 90.0, 90.0 |
| Resolution (Å) | 50.00–3.50 (3.56–3.50) | 50.00–3.23 (3.29–3.23) | 50.00–1.80 (1.83–1.80) |
| $R_{merge}$ | 0.298 (0.980) | 0.344 (0.958) | 0.103 (0.929) |
| I/σI | 4.8 (1.1) | 4.3 (1.6) | 22.0 (2.5) |
| Completeness (%) | 98.8 (96.5) | 99.9 (99.9) | 97.7 (95.8) |
| Redundancy | 4.5 (3.5) | 7.5 (6.4) | 10.1 (9.5) |
| Refinement | | | |
| Resolution (Å) | 3.69 | 3.39 | 1.80 |
| No. reflections | 36,773 | 41,321 | 44,296 |
| $R_{work}$/$R_{free}$ | 0.2798/0.3151 | 0.3041/0.3281 | 0.2120/0.2334 |
| *No. atoms* | | | |
| Protein | 22,873 | 18,763 | 3657 |
| Water | 180 | 386 | 212 |
| *B-factors (Å$^2$)* | | | |
| Protein | 95.5 | 125.6 | 23.3 |
| Water | 33.2 | 55.9 | 28.5 |
| *R.m.s. deviations* | | | |
| Bond length (Å) | 0.008 | 0.011 | 0.015 |
| Bond angles (°) | 1.516 | 1.785 | 1.500 |
| *Ramachandran plot* | | | |
| Favored region | 94.96 | 95.72 | 97.98 |
| Allowed region | 4.86 | 4.28 | 2.02 |
| Outlier region | 0.18 | 0.00 | 0.00 |

*Highest resolution shell is shown in parentheses.

better understand the differences in the mechanisms of Grc3 activation of Las1 between different species, we also solved the cryo-EM and crystal structures of full-length CjLas1-Grc3 complex at 3.39 Å and 3.39 Å resolution (*Figure 3*, *Figure 3—figure supplement 1*, *Tables 1 and 2*), respectively. To be noted, the crystal structures were solved by molecular replacement method with the cryo-EM maps of Las1-Grc3 complexes. Since the crystal structures have similar quality and resolve more structural information than the cryo-EM structures (*Figure 3—figure supplement 2*), the subsequent presentations and descriptions are based more on the crystal structures.

Both structures reveal that Las1 and Grc3 assemble into a tetramer with two copies of each (*Figures 2B and C and 3B and C*). In the tetramer, one Las1 and Grc3 molecules form a C2 symmetry with another Las1 and Grc3 molecules. Las1 consists of a relatively conserved N-terminal HEPN domain, a poorly conserved central CC domain, and a short C-terminal tail motif (LCT) (*Figures 2A and 3A*, *Figure 3—figure supplement 3*). Grc3 is composed of an N-terminal domain (NTD), a central polynucleotide kinase (PNK) domain, a C-terminal domain (CTD), and a short C-terminal loop motif (GCT). A CC domain of Las1 and a copy of Grc3 are assembled into an architecture similar to one wing

**Table 2.** Cryo-electron microscopy (cryo-EM) data collection, refinement, and validation statistics.

| | ScLas1-Grc3 | CjLas1-Grc3 |
|---|---|---|
| **Data collection and processing** | | |
| Microscope | Titan Krios | Titan Krios |
| Voltage (kV) | 300 | 300 |
| Camera | Gatan K3 | Gatan K3 |
| Magnification | 105,000× | 105,000× |
| Pixel size (Å) | 0.82 | 0.82 |
| Total exposure (e-/Å$^2$) | 50 | 50 |
| Exposure time (s) | 3 | 3 |
| Number of frames per exposure | 30 | 30 |
| Energy filter slit width (keV) | 20 | 20 |
| Data collection software | EPU | EPU |
| Defocus range (μm) | −1.3 to −2.7 | −1.2 to −3 |
| Number of micrographs | 2520 | 8616 |
| Number of initial particles | 525,213 | 2,215,555 |
| Symmetry | C2 | C2 |
| Number of final particles | 264,341 | 523,843 |
| Resolution (0.143 gold standard FSC, Å) | 3.07 | 3.39 |
| Local resolution range (Å) | 2.8–4.8 | 2.8–4.8 |
| Microscope | Titan Krios | Titan Krios |
| **Refinement** | | |
| *Model composition* | | |
| Nonhydrogen atoms | 11,212 | 10,013 |
| Protein residues | 1426 | 1339 |
| *B*-factors (Å$^2$) | | |
| Protein | 92.58 | 85.58 |
| R.m.s. deviations | | |
| Bond length (Å) | 0.009 | 0.004 |
| Bond angles (°) | 1.055 | 0.849 |
| Validation | | |
| MolProbity score | 2.90 | 2.69 |
| Clashscore | 20.56 | 18.60 |
| Rotamer outliers (%) | 8.75 | 7.86 |
| Ramachandran plot | | |
| Favored region | 94.46 | 96.38 |
| Allowed region | 4.83 | 3.31 |
| Outlier region | 0.71 | 0.31 |

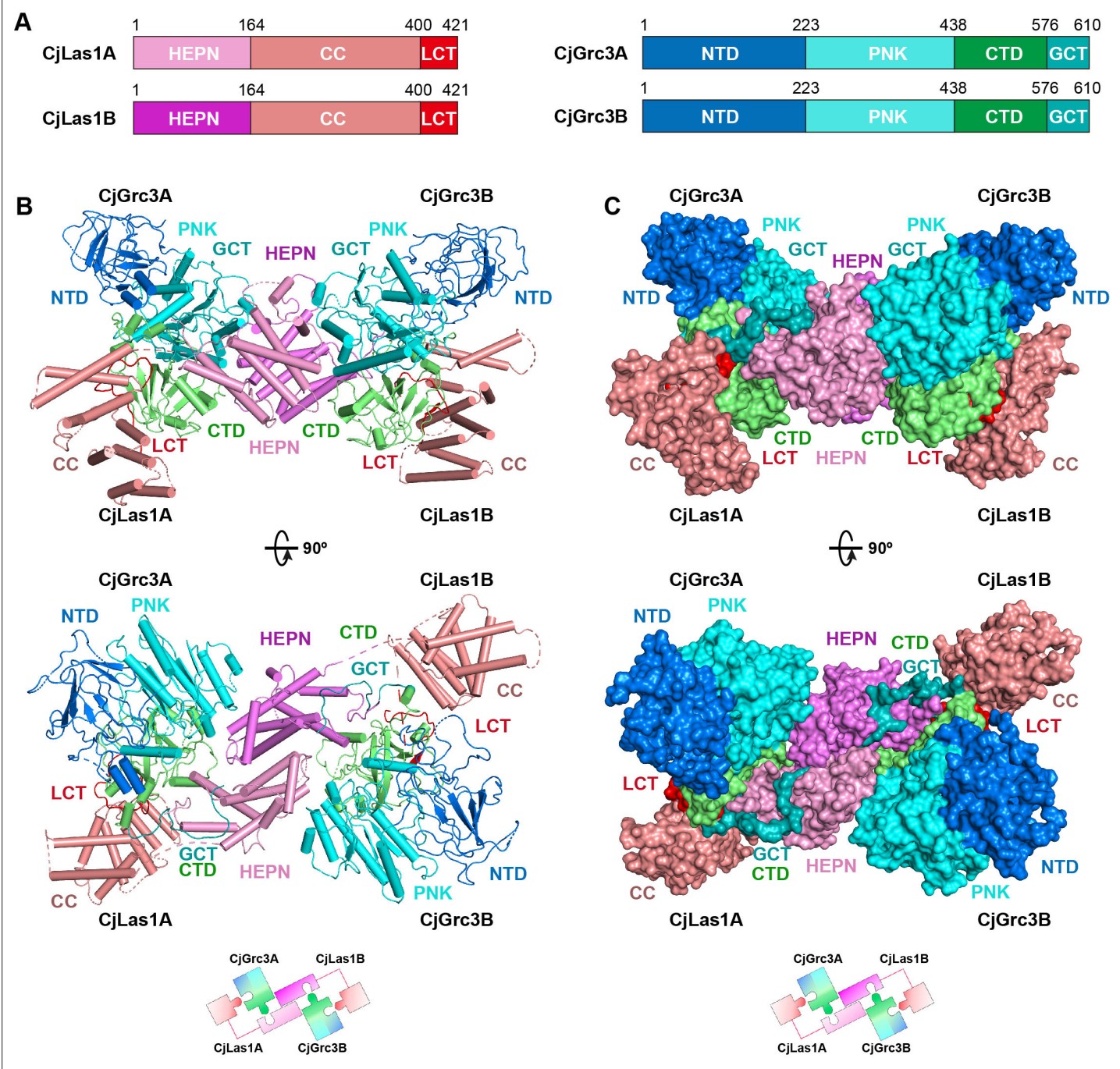

**Figure 3.** Overall structure of CjLas1-Grc3 complex. (**A**) Domain organization of CjLas1 and CjGrc3. (**B**) Top: ribbon representations of CjLas1-Grc3 complex. Color coding used for Las1 and Grc3 is identical to that used in (**A**). Bottom: a diagram showing how the Las1-Grc3 tetramer is formed. (**C**) Top: surface representations of CjLas1-Grc3 complex. Color coding used for Las1 and Grc3 is identical to that used in (**A**). Bottom: a diagram showing how the Las1-Grc3 tetramer is formed.

The online version of this article includes the following figure supplement(s) for figure 3:

**Figure supplement 1.** Single-particle cryo-electron microscopy (cryo-EM) analysis of the CjLas1-Grc3 complex.

**Figure supplement 2.** Comparison of cryo-electron microscopy (cryo-EM) and crystal structures of Las1-Grc3 complexes.

**Figure supplement 3.** Sequence alignment of Las1 proteins.

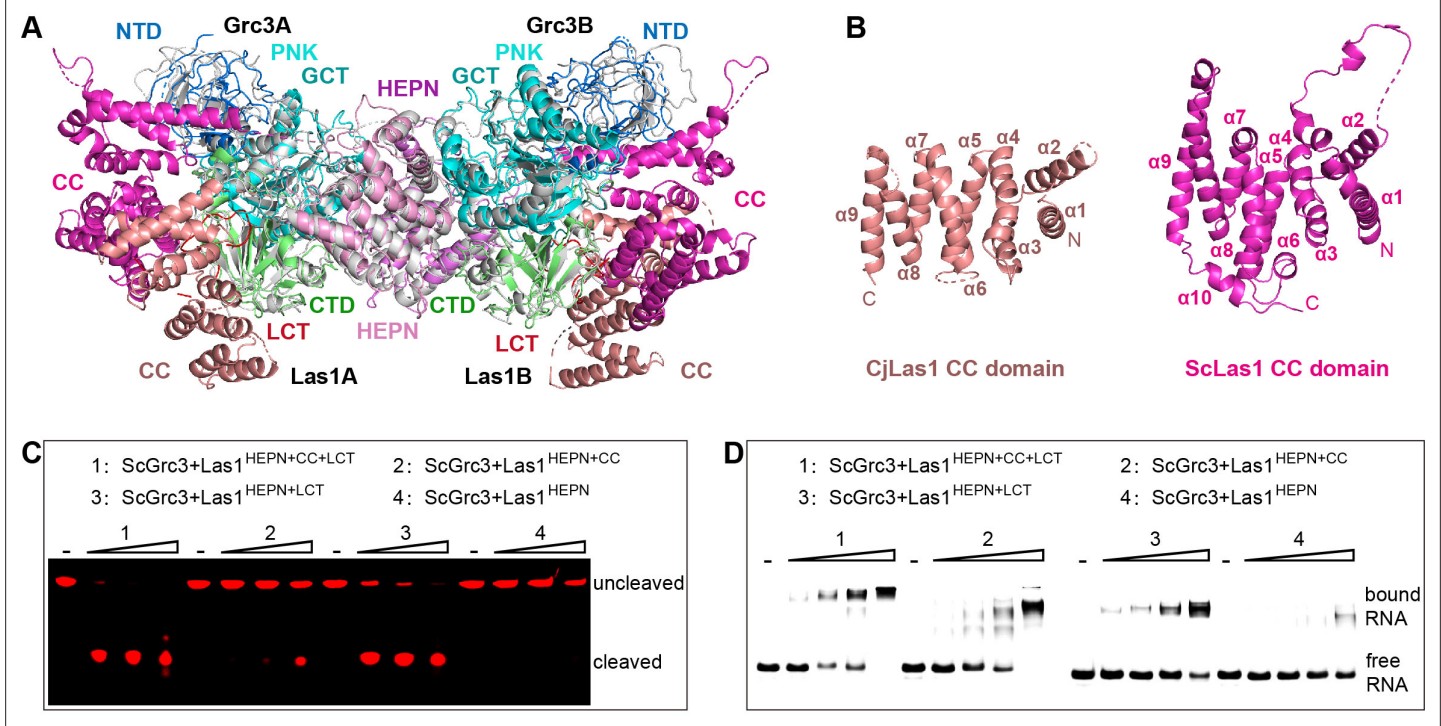

**Figure 4.** The coiled-coil (CC) domain contributes to ITS2 RNA binding and cleavage. (**A**) Structural comparison between ScLas1-Grc3 complex and CjLas1-Grc3 complex. Color coding used for ScLas1 and ScGrc3 is identical to that used in *Figure 2A*. The CC domain of CjLas1 is colored in salmon, other domains of CjLas1 and all domains of CjGrc3 are colored in gray. (**B**) Structures of CjLas1 CC domain (in salmon) and ScLas1 CC domain (in light magenta). (**C**) In vitro RNA cleavage assay using indicated truncations of ScLas1. HEPN: residues 1–165; CC: residues 181–430; LCT: residues 430–502. (**D**) Electrophoretic mobility shift assay using indicated truncations of ScLas1. All experiments were repeated three times.

The online version of this article includes the following source data for figure 4:

**Source data 1.** Original files for the RNA cleavage and binding analysis in *Figure 4C and D*.

**Source data 2.** Original scans of the relevant RNA cleavage and binding analysis in *Figure 4C and D* with band and sample labels.

of a butterfly. The NTD and PNK domains of Grc3 constitute the forewing, while the CTD domain of Grc3 and the CC domain of Las1 form the hindwing. All domains are perfectly stacked together to build a compact and stable tetramer architecture (*Figures 2C and 3C*). Two HEPN domains from two Las1 copies are tightly symmetrically stacked together to form a HEPN dimer. Two Grc3 molecules are assembled on both sides of Las1 HEPN domains to stabilize the conformation of HEPN dimer. The CC domain is located on one side of Grc3 and forms a sandwich-shaped structure with the Grc3 and HEPN domains (*Figures 2B and 3B*), resulting in Grc3 being anchored by the CC and HEPN domains. In addition, the CTD of Grc3 and the HEPN domain of Las1 tightly grasp Las1 LCT and Grc3 GCT, respectively, making Las1-Grc3 tetramer assembly more stable.

## Las1 CC domain contributes to ITS2 binding and enhances cutting

In order to explore whether there are structural differences between ScLas1-Grc3 and CjLas1-Grc3 complexes, we conducted structural comparison by superposition of the structures of the two complexes. Structural superposition shows that the conformations of all Grc3 domains and Las1 HEPN domains are almost identical in the two complexes, while the conformations of Las1 CC domains are significantly different (*Figure 4A*).

Dali server reveals that the CC domain shares little structural similarity with any known proteins (*Holm and Rosenström, 2010*). The CC domain is mainly composed of α helices and is the largest domain of Las1 (*Figure 4B*), with low-sequence similarity among different species (*Figure 3—figure supplement 3*), yet its function remains unknown. To determine whether it plays a role in Las1 catalyzing ITS2 cleavage, we performed electrophoretic mobility shift assays (EMSA) and in vitro RNA cleavage assays using Las1 proteins with or without CC domain truncations. Our data shows that the

CC domain contributes to the binding of ITS2 RNA and then facilitates ITS2 cleavage (*Figure 4C and D*), suggesting that the CC domain may play a role of ITS2 stabilization in the Las1 cutting reaction.

## Grc3 GCT binds to the HEPN active center and mediates Las1 activation

The ScLas1-Grc3 tetramer structure shows that two Grc3 GCT are stabilized by two HEPN domains of Las1 (*Figure 2*). Each Grc3 GCT binds to a groove in each Las1 HEPN domain and extends to the ribonuclease active center (*Figure 5A*, *Figure 5—figure supplement 1*), which is formed by the two HEPN domains via dimerization. Both the crystal and cryo-EM structures show that the conserved catalytic residues Arg129, His130, His134 with two copies from two HEPN domains form a symmetric catalytic active pocket (*Figure 5B*, *Figure 5—figure supplement 2*). The two C-terminals of Grc3 GCTs tightly bind to the catalytic active pocket through packing and hydrogen bond interactions (*Figure 5—figure supplement 3*). Specifically, the side chain of Trp617 within Grc3 GCT inserts into the active pocket and forms packing and hydrogen bond with catalytic residue His134 (*Figure 5B*). Moreover, Trp617 and His615 in Grc3 form extensive hydrogen bond with residues Arg136, Leu99, Gly98, and His54 within Las1. The Trp617 residue is highly conserved in Grc3 (*Figure 5C*), and its mutation completely abolishes the ITS2 cleavage (*Figure 5D*), but has little effect on Las1 binding (*Figure 5E*), revealing that it plays a crucial role in the activation of HEPN endonuclease. We also mutated each of the residues at the C-terminal of Grc3 and examined the nuclease activity of Las1 complexes with these mutants. Our data shows that alanine substitution of multiply conserved residues dramatically reduces the ITS2 cleavage (*Figure 5F*, *Figure 5—figure supplement 4*), indicating that these residues are also essential for coordinating Las1 HEPN endonuclease activation.

## Las1 LCT drives Las1-Grc3 complex assembly

Our crystal and cryo-EM structures exhibit that the Las1 LCT is located in a groove in the CTD domain of Grc3 (*Figure 6A*, *Figure 6—figure supplement 1*). Structure analysis reveals that the Las1 LCT is stabilized by the Grc3 CTD through extensive hydrophobic interactions and hydrogen bonding. The side chains of Trp488, Trp494, and Phe499 of Las1 LCT are inserted into the three hydrophobic core regions of Grc3 CTD and form stable hydrophobic interactions with multiple hydrophobic residues of Grc3 (*Figure 6B and C*, *Figure 6—figure supplement 1*). Additionally, the side chain of Asn487 and the main chains of Lys497 and Ser489 of Las1 LCT form multiple hydrogen bonds with the sides of Trp573, His543 and the main side of Gln468 of Grc3 (*Figure 6B and C*, *Figure 6—figure supplement 1*). Mutations and deletions of Las1 LCT reduce the enzyme activity of Las1 (*Figure 6—figure supplement 2*). LCT deletions also significantly affect the association between Las1 and Grc3 (*Figure 6—figure supplement 3*). These results highlight the functional significance of the sequence-dependent recognition of Las1 LCT by Grc3 CTD.

## Special crystal structure of Las1 HEPN domain

The biochemical data shows that Las1 exhibits very weak ability to cut ITS2 RNA at both C2 and C2′ sites in the absence of Grc3 (*Figure 1B*, *Figure 1—figure supplement 1*), suggesting that Las1 may exist in a low-activity conformation state prior to assembly with Grc3 (*Pillon et al., 2017*). To determine the structure of Las1 in the low-activity conformation, we screened about 1200 crystallization conditions with full-length Las1 proteins. Unfortunately, we did not obtain any crystals. We conjectured that the CC domain of Las1 might have a flexible conformation in the absence of Grc3, and then attempted to crystallize Las1 truncated by the CC domain and LCT. After screening a large number of crystallization conditions, we successfully obtained well-ordered crystals of the CjLas1 HEPN domain and determined its structure at 1.80 Å resolution using molecular replacement method (*Figure 7A*, *Table 1*).

The crystallographic asymmetric unit contained three Las1 HEPN molecules (HEPN1, HEPN2, and HEPN3), each of which assumes an all α-helical fold (*Figure 7A*). Analysis of crystal packing and interactions across HEPN-HEPN interfaces suggests that the biological unit of Las1 HEPN domain may contain a homodimer and a monomer. HEPN1 and HEPN2 molecules form a face-to-face dimer related through the preudo-twofold axis, similar to the HEPN dimer in the Las1-Grc3 tetramer structure. Most notably, the molecule HEPN3 is likely not to form a dimer similar to HEPN1-HEPN2 with its symmetry-related molecule. Firstly, the structural superposition of HEPN3 and its symmetry-related

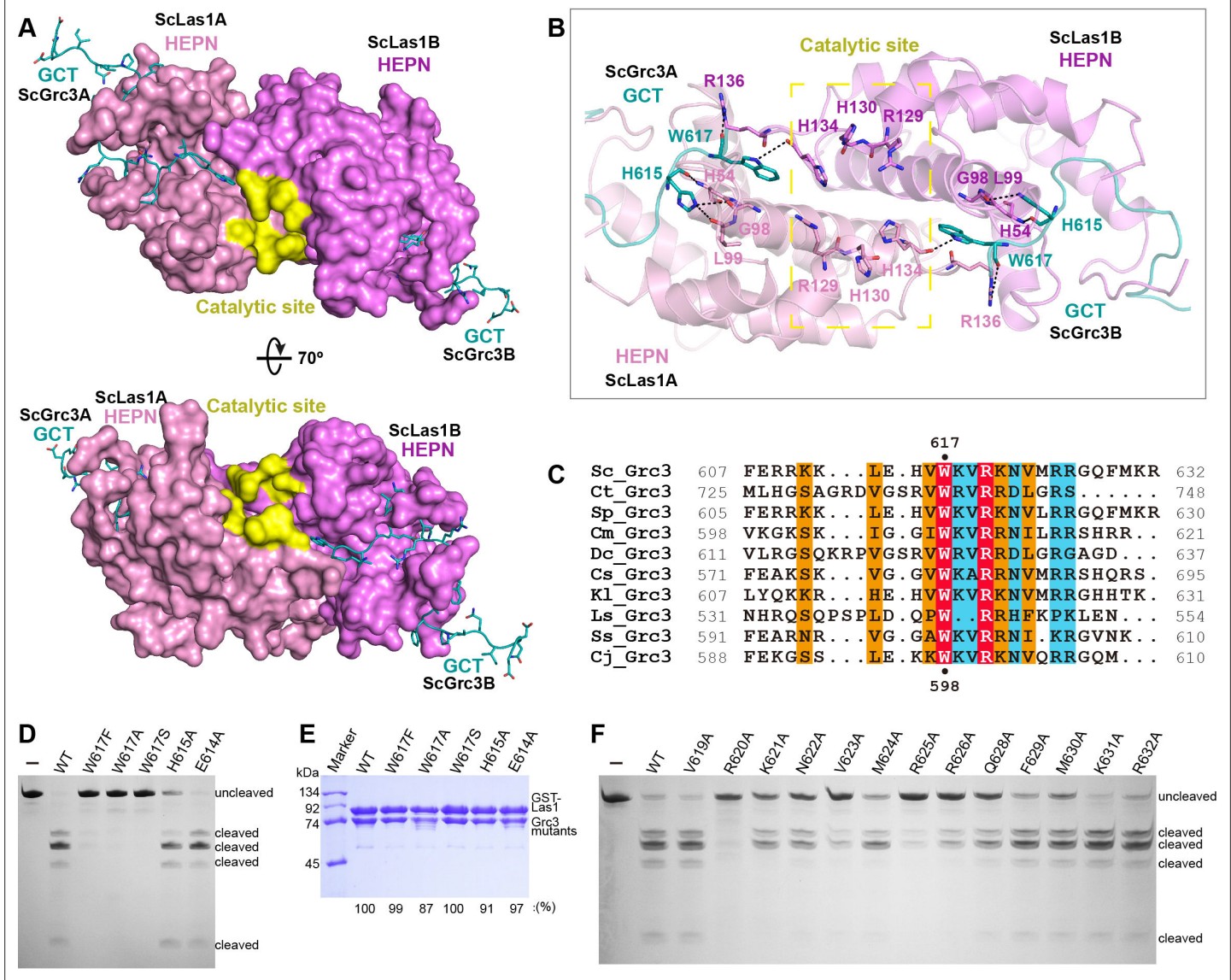

**Figure 5.** ScGrc3 GCT mediates the ITS2 cleavage activity of ScLas1. (**A**) The crystal structure shows that ScGrc3 GCT binds at an active channel of ScLas1 HEPN dimer. Two HEPN domains of Las1 are colored in pink and violet, respectively. GCTs of Grc3 are colored in teal. The catalytic site is highlighted in yellow. HEPN domains are shown as surfaces, while GCTs are shown as sticks. (**B**) Detailed interactions between ScGrc3 GCT and ScLas1 HEPN domain. (**C**) Sequence alignments of Grc3 GCTs. Identical residues are highlighted in red. Basically constant residuals are shaded in blue. Conserved residues are shaded in orange. (**D**) In vitro enzymatic assay of mutations of ScGrc3 residues Glu614, His615, and Trp617. (**E**) GST pull-down experiment assaying the ability of ScGrc3 mutants to interact with ScLas1. (**F**) In vitro enzymatic assay of alanine mutations of ScGrc3 C-terminal residues. All experiments were repeated three times.

The online version of this article includes the following source data and figure supplement(s) for figure 5:

**Source data 1.** Original files for enzymatic assay and GST pull-down analysis in *Figure 5D–F*.

**Source data 2.** Original scans of the relevant enzymatic assay and GST pull-down analysis in *Figure 5D–F* with band and sample labels.

**Figure supplement 1.** Cryo-electron microscopy (cryo-EM) data shows the interactions between ScGrc3 GCTs and ScLas1 HEPN domains.

**Figure supplement 2.** Comparison of crystal structure and cryo-electron microscopy (cryo-EM) structure of HEPN domains and GCTs in ScLas1-Grc3 complex.

**Figure supplement 3.** Cryo-electron microscopy (cryo-EM) data (top) and crystal data (bottom) show the electron density of the catalytic site of ScLas1 and the GCTs of ScGrc3.

**Figure supplement 4.** In vitro enzymatic assay of alanine mutations of conserved CjGrc3 residues Trp618, Arg601, Arg606, and Arg607.

**Figure supplement 4—source data 1.** Original files for the RNA cleavage analysis in *Figure 5—figure supplement 4*.

*Figure 5 continued on next page*

*Figure 5 continued*

**Figure supplement 4—source data 2.** Original scans of the relevant RNA cleavage analysis in *Figure 5—figure supplement 4* with band and sample labels.

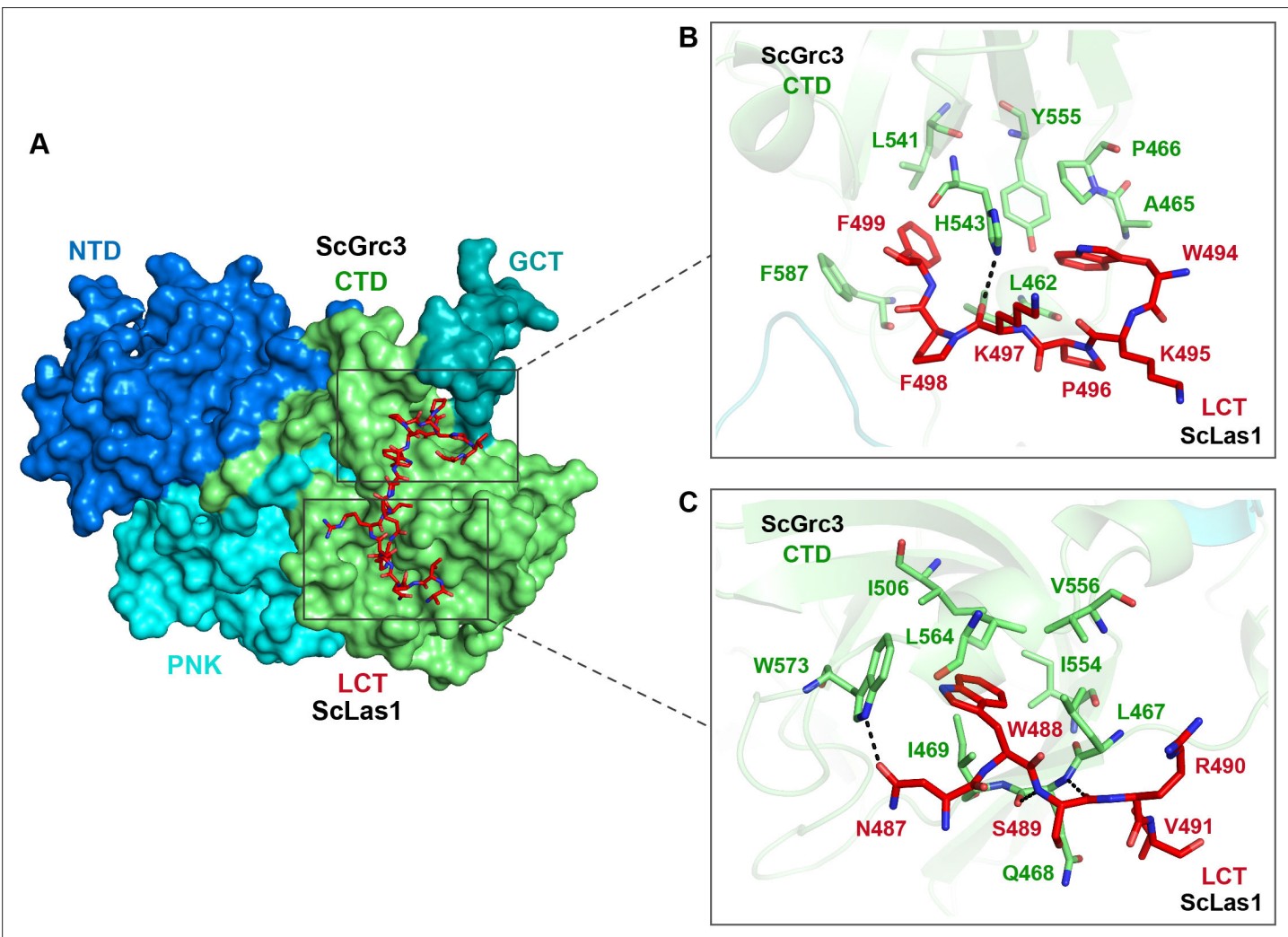

**Figure 6.** Las1 LCT drives Las1-Grc3 cross-talk. (**A**) The crystal structure shows that ScLas1 LCT binds to the CTD domain of ScGrc3. ScGrc3 is shown as surface, ScLas1 LCT is shown as stick. (**B**) Detailed interactions between C-terminal residues of ScLas1 LCT and ScGrc3 CTD domain. (**C**) Detailed interactions between N-terminal residues of ScLas1 LCT and ScGrc3 CTD domain.

The online version of this article includes the following source data and figure supplement(s) for figure 6:

**Figure supplement 1.** Cryo-electron microscopy (cryo-EM) data shows the interactions between ScLas1 LCT and ScGrc3 CTD domain.

**Figure supplement 2.** Denaturing gel showing the ITS2 pre-RNA cleavage by mutation or deletion of the interacting residues of ScLas1 LCT.

**Figure supplement 2—source data 1.** Original file for the RNA cleavage analysis in *Figure 6—figure supplement 2*.

**Figure supplement 2—source data 2.** Original scan of the relevant RNA cleavage analysis in *Figure 6—figure supplement 2* with band and sample labels.

**Figure supplement 3.** GST pull-down experiment assaying the Grc3 binding ability by mutation or deletion of the interacting residues of ScLas1 LCT.

**Figure supplement 3—source data 1.** Original file for the GST pull-down analysis in *Figure 6—figure supplement 3*.

**Figure supplement 3—source data 2.** Original scan of the relevant GST pull-down analysis in *Figure 6—figure supplement 3* with band and sample labels.

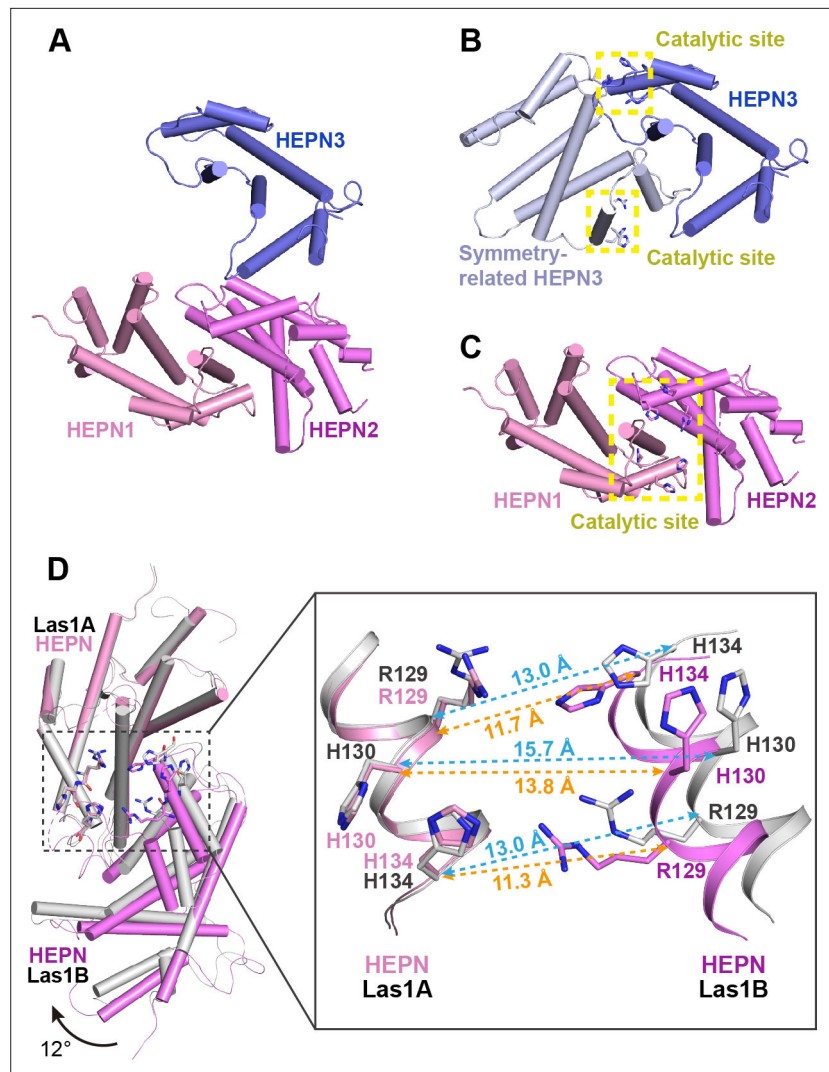

**Figure 7.** Activation mechanism of Las1 by Grc3. (**A**) Crystal structure of CjLas1 HEPN domain. (**B**) The HEPN3 molecule (in slate) and its symmetry-related molecule (in blue white) in Las1 HEPN domain structure. (**C**) The HEPN1 (in pink) and HEPN2 (in violet) molecules in Las1 HEPN domain structure. (**D**) Structural comparison of HEPN dimers between CjLas1-Grc3 complex (Las1A HEPN in pink, Las1B HEPN in violet) and CjLas1 HEPN domain (in gray). Inset: a magnified view of the comparison of the catalytic site in the two structures.

The online version of this article includes the following figure supplement(s) for figure 7:

**Figure supplement 1.** Structural superposition of HEPN3 (in slate) and its symmetry-related molecule (in blue white) with the HEPN1(in pink)-HEPN2 (in violet) dimer.

**Figure supplement 2.** Structural comparison of HEPN dimers between ScLas1-Grc3 complex (Las1A HEPN in pink, Las1B HEPN in violet) and CjLas1 HEPN domain (in gray).

**Figure supplement 3.** Structural comparison of catalytic sites between Las1-Grc3 complexes.

molecule with the HEPN1-HEPN2 dimer shows that HEPN1 and HEPN3 molecules superimpose well with core root mean-square deviation (RMSD) of 0.7 Å for 133 Cα atoms, while HEPN2 and symmetry-related HEPN3 molecule exhibit significantly different conformations (*Figure 7—figure supplement 1*). Secondly, the catalytic residues in HEPN1 and HEPN2 domains are close to each other to form a compact active center, whereas the catalytic residues in HEPN3 and its symmetry-related molecule remain far apart, presenting two separate catalytic sites (*Figure 7B and C*). These observations suggest that the HEPN3 molecule in the asymmetric unit is probably a monomer, which forms a special packing with the dimer of HEPN1-HEPN2 in the crystal.

Together, these results indicate a monomer–dimer equilibrium in the Las1 HEPN domain, which is consistent with the previously reported SEC-MALS data (*Pillon et al., 2017*).

## Conformational changes in Las1 HEPN domain upon Grc3 binding

Since Grc3 binding significantly activates the endonuclease activity of Las1, the HEPN nuclease domains probably undergo conformational changes before and after Grc3 binding. We compared the HEPN domain dimer structures in CjLas1 and CjLas1-Grc3 complex by superposition and observed that significant conformational changes occur in HEPN domains upon Grc3 binding (*Figure 7D*). The conformational change is ~3–4 Å Cα RMSD across all ~150 residues in the domain (~90 residues forming a stable core that only changes by ~1 Å). There is also a shift in the associated HEPN domain in Las1B domain compared to the bound HEPN in the Las1-Grc3 complex, as shown in *Figure 7D*: ~1 Å shift and ~12° rotation. In order to investigate whether conformational changes occurred in the catalytic center, we further compared the catalytic pocket within the HEPN domains of the two structures. Remarkably, large conformational changes are observed in the catalytic pocket (*Figure 7D*). In the Grc3-free Las1 structure, the catalytic residues Arg129, His130, and His134 within one HEPN domain are far from the catalytic residues in the other HEPN domain, while in the Grc3-bound structure, the catalytic residues within two HEPN domains are close to each other (*Figure 7D*, *Figure 7—figure supplement 2*). These observations indicate that the Grc3 binding not only stabilizes the Las1 HEPN dimer, but also promotes the formation of a more compact catalytic pocket, showing better catalytic activity for specific cleavage of ITS2. Notably, compared with CjGrc3-bound CjLas1, ScGrc3-bound ScLas1 has a more compact catalytic pocket (*Figure 7D*, *Figure 7—figure supplement 2*), which may explain why ScLas1 shows better ITS2 cleavage activity than CjLas1 in the presence of Grc3 (*Figure 1B*, *Figure 1—figure supplement 1*). In addition, His130 in the ScLas1-Grc3 complex active site and the analogous His130 in the

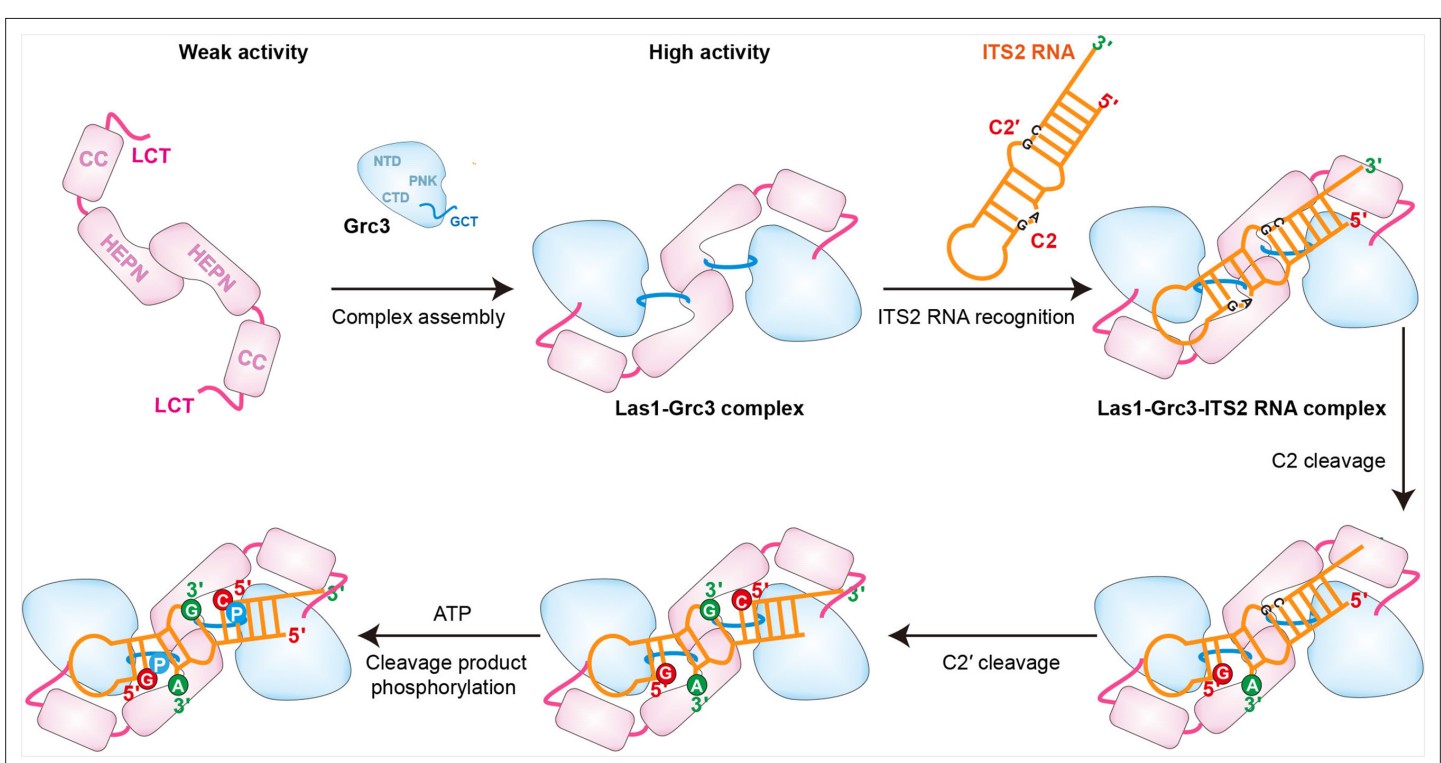

**Figure 8.** Model for Grc3-mediated Las1-catalyzed ITS2 pre-rRNA processing. Prior to assembly with Grc3, Las1 shows weak processing activity for ITS2 precursor RNA. When combined with Grc3 to form a tetramer complex, Las1 shows high processing activity for ITS2 precursor RNA. Las1 specifically cleaves ITS2 at the C2 and C2' sites to generate 5'-OH terminus products. The 5'-OH terminus products are further phosphorylated by Grc3 when in the presence of ATP.

The online version of this article includes the following figure supplement(s) for figure 8:

**Figure supplement 1.** Comparison of Las1-Grc3, Ire1, and RNase L.

CjLas1-Grc3 complex active site adopt two different conformations (*Figure 7—figure supplement 3*), which are similar to the conformations of His142 of CtGrc3-bound CtLas1 in two different active states (*Pillon et al., 2019*). This observation may also provide a structural rationale for the activation of ScLas1 and CtLas1 by Grc3, as well as for the fairly weak activation of CjLas1 by CjGrc3. Of course, since the structure of Las1 analyzed by us lacks the LCT and CC domains, we do not know whether these two domains have an effect on the conformational changes of HEPN domains before and after Grc3 binding.

## Discussion

Removal of the ITS2 is the requirement for the maturation of 5.8S rRNA and 25S rRNA, which is an important step during the eukaryotic 60S subunit synthesis. In this study, the structural, biochemical, and functional analysis of ITS2 RNA processing machinery provides a critical step toward understanding the molecular mechanism of Grc3-activated ITS2 processing by Las1 endoribonuclease.

### Mechanism of ITS2 processing by Las1-Grc3 dual enzyme complex

Based on our findings, we propose a model of ITS2 processing by Las1 endoribonuclease and its activator Grc3 kinase (*Figure 8*). In the absence of Grc3, Las1 has very weak ITS2 cleavage activity due to its unstable HEPN dimer. However, in the presence of Grc3, two copies of Las1 and Grc3 assemble into a dynamic tetramer that shows high activity to cleavage ITS2. Las1 initially processes ITS2 at the C2 site, generating a 5.8S rRNA precursor with 2',3'-cyclic phosphate and a 25S rRNA precursor with 5'-hydroxyl end (*Figures 1I and 8*). The second step of ITS2-specific cleavage occurs at the C2' site to remove the 5'-region of the 25S rRNA precursor. The C2' cleavage also produces a 5'-hydroxyl product, which is rapidly phosphorylated by Grc3 only in the presence of ATP. After phosphorylation, the 25S precursor is further processed by a Rat1-Rai1 complex with 5'−3' exonuclease activity, which degrades the 5'-region of the 25S rRNA precursor and subsequently generate mature 25S rRNA (*Figure 1I*; *Gasse et al., 2015*). Furthermore, the nuclear exosome complex drives the maturation of the 5.8S RNA by removing the 3'-end of the precursor through its 3'−5' exonuclease activity (*Fromm et al., 2017*).

In addition, based on the structural data, our model may also provide some insight into how Las1-Grc3 complex cleaves ITS2 RNA at both C2 and C2' positions. The Las1-Grc3 tetramer complex has one nuclease active center and two kinase active centers. The nuclease active center consists of two Las1 molecules in a symmetric manner, while the kinase active center consists of only one Grc3 molecule. The ITS2 RNA is predicted to form a stem-loop structure. The symmetrical nuclease active center recognizes the stem region of ITS2 RNA and makes it easy to perform C2 and C2' cleavages on both sides of the stem. C2 and C2' cleavage products are further phosphorylated by two Grc3 kinase active centers, respectively.

### Distinct activation mechanism for Las1 HEPN nuclease

Structural and biochemical results highlight the mechanism of Grc3 GCT-mediated Las1 HEPN nuclease activation, which is quite distinct from other HEPN nucleases such as CRISPR-Cas and toxin-antitoxin-associated HEPN RNases, Cas13a, Csm6, RnlA. Though proper dimerization of the HEPN domain is critical for HEPN nuclease activity, other factors are also common requirements for HEPN activation. For example, activation of the Cas13a HEPN enzyme requires target RNA binding and guide-target RNA duplex formation, while allosterically activation of the Csm6 HEPN nuclease is dependent on a cyclic oligoadenylate (*Liu et al., 2017b*; *Niewoehner et al., 2017*). In addition, stimulating the toxicity of RnlA HEPN RNase requires an association between RnlA and RNase HI, and trigging activation of the HEPN domains of Ire1 and RNase L needs binding of ATP to their kinase domains (*Lee et al., 2008*; *Naka et al., 2014*; *Wang et al., 2014*). It is noteworthy that these activators do not interact directly with the HEPN domain, but bind to other domains to induce a conformational transition of the HEPN domain from inactive to active state. In contrast, Grc3 GCTs bind directly to the active center of HEPN domains and form hydrogen bonds with catalytic residues, which appears to provoke rearrangements of the active site required for the activation of HEPN. Regulation of HEPN RNases by interacting with catalytic residues may be a direct and effective measure.

## Comparison of Las1-Grc3 complex and other nuclease-kinase machines

Ire1 and RNase L are also nuclease-kinase machines that contain a HEPN domain and a protein kinase domain or a pseudo-protein kinase domain, playing a fundamental role in RNA degradation related to a variety of cellular processes. Although Ire1 and RNase L also require higher-order assembly during RNA degradation, the Las1-Grc3 complex shows distinct structure assembly with them (*Figure 8— figure supplement 1*). Both the kinase domain and the HEPN domain in Ire1 and RNase L adopt a parallel back-to-back dimer configuration (*Lee et al., 2008*; *Wang et al., 2014*). In Las1, only the HEPN domain forms a dimer architecture, and Grc3 does not contact each other, but binds on both sides to stabilize the HEPN dimer. An ankyrin repeat domain in RNase L and an endoplasmic reticulum (ER) luminal domain in Ire1 have been proposed to promote dimerization of the kinase and HEPN domains (*Credle et al., 2005*; *Huang et al., 2014*). Whereas the additional CC domain in Las1 is likely to contribute to RNA binding and facilitate cleavage (*Figure 4C and D*). In addition, Las1, Ire1, and RNase L recognize similar RNA cleavage motifs despite being involved in different RNA processing, splicing, and degradation pathways. Cleavage studies show a preference of Las1 for UAG and UGC motifs, which are also the universal cleavage motifs observed in mRNA decay by Ire1 and RNase L (*Pillon and Stanley, 2018*).

## Materials and methods

### Protein expression and purification

For Las1-Grc3 complex expression and purification, *Saccharomyces cerevisiae Las1* (*ScLas1*) *gene* was cloned into pET23a vector with an N-terminal His$_6$-tag and *Cyberlindnera jadinii* (*CjLas1*) gene was cloned into pET28a vector with an N-terminal His$_6$SUMO-tag, while *ScGrc3* and *CjGrc3* genes were cloned into modified pET28a vector containing an N-terminal SUMO-tag, followed by a ubiquitin-like protein1 (Ulp1) protease cleavage site. All recombinant plasmids were transformed into *Escherichia coli* Rosetta (DE3) (Novagen) cells and grown in LB broth at 37°C for 3 hr. After culturing to an OD$_{600}$ of 0.6–0.8, protein expression was induced with 0.2 mM isopropyl-1-thio-β-D-galactopyranoside (IPTG) at 16°C for 14 hr. For the purification of ScLas1-Grc3 complex proteins, cells expressing ScLas1 proteins and ScGrc3 proteins were collected and co-lysed by sonication in a buffer containing 20 mM Tris–HCl, pH 7.5, 500 mM NaCl. After centrifugation, the supernatant was incubated with Ni Sepharose (GE Healthcare), and the bound protein was eluted with buffer containing 20 mM Tris–HCl, pH 7.5, 300 mM NaCl, 300 mM imidazole. Eluted protein was digested with Ulp1 protease at 4°C for 2 hr and then further purified on a Heparin HP column (GE Healthcare), eluting with a linear gradient of increasing NaCl concentration from 300 mM to 1 M in 20 mM Tris–HCl, pH 7.5 buffer. The fractions containing the protein of interest were concentrated and further purified by size-exclusion chromatography (Superdex 200 Increase 10/300, GE Healthcare) in a buffer containing 20 mM Tris–HCl, pH 7.5, 300 mM NaCl, and 1 mM Tris(2-carboxyethyl)phosphine hydrochloride (TCEP). Proteins were collected and concentrated to a final concentration of 15 mg/ml. The mutants and truncations of ScLas1 and ScGrc3 were purified with an identical protocol.

For the purification of CjLas1-Grc3 complex proteins, cells expressing CjLas1 proteins and CjGrc3 proteins were collected and co-lysed by sonication in a buffer containing 20 mM Tris–HCl, pH 7.5, 500 mM NaCl. After centrifugation, the supernatant was incubated with Ni Sepharose (GE Healthcare), and the bound protein was eluted with buffer containing 20 mM Tris–HCl, pH 7.5, 350 mM NaCl, 300 mM imidazole. Eluted protein was digested with Ulp1 protease at 4°C for 2 hr and then further purified on a Heparin HP column (GE Healthcare), eluting with buffer containing 20 mM Tris–HCl, pH 7.5, 1 M NaCl. The protein was further purified by size-exclusion chromatography (Superdex 200 Increase 10/300, GE Healthcare) in a buffer containing 20 mM Tris–HCl, pH 7.5, 350 mM NaCl, and 1 mM TCEP. Proteins were collected and concentrated to a final concentration of 15 mg/ml. Mutants were expressed and purified as wild-type protein. The CjLas1 HEPN construct was purified with an identical protocol.

For the experiment of GST pull-down, the wild-type *ScLas1* gene and its mutants were cloned into a modified pET23a vector (Novagen) with an N-terminal His$_6$GST-tag, while the wild-type *ScGrc3* gene and its mutants were cloned into a modified pET28a vector containing an N-terminal His$_6$SUMO-tag. Proteins were overexpressed in *E. coli* Rosetta (DE3) (Novagen) cells and cultured in LB broth to an OD$_{600}$ of 0.6–0.8. Then the target proteins were induced with 0.2 mM IPTG and grown

for an additional 14 hr at 16°C. Cells were collected and lysed by sonication in a buffer containing 20 mM Tris–HCl, pH 7.5, 500 mM NaCl. After centrifugation, the supernatant was incubated with Ni Sepharose (GE Healthcare), and the bound protein was eluted with buffer containing 20 mM Tris–HCl, pH 7.5, 300 mM NaCl, 300 mM imidazole. The GST-ScLas1 mutants were further purified on a Heparin HP column (GE Healthcare), and ScGrc3 mutants were purified on a HitTrap Q HP column (GE Healthcare). All proteins were further purified by size-exclusion chromatography (Superdex 200 Increase 10/300, GE Healthcare) in a buffer containing 20 mM Tris–HCl, pH 7.5, 300 mM NaCl.

## Crystallization, data collection, and structure determination

Crystals of ScLas1-Grc3 complex, CjLas1-Grc3 complex, and CjLas1 truncated protein (HEPN domain) were first obtained using the sitting drop vapor diffusion method using high-throughput crystallization screening kits (Hampton Research, Molecular Dimensions, and QIAGEN). Crystals were then grown in a mixed solution containing 1 μl complex solution and 1 μl of reservoir solution using the hanging drop vapor diffusion method at 16°C. For growing large crystals, crystals were further optimized by using seeding technique. Well-diffracting crystals of ScLas1-Grc3 complex were grown in a reservoir solution containing 2% (v/v) tacsimate, pH 7.0, 0.1 M imidazole, pH 7.0, 2% (v/v) 2-propanol, and 9% (w/v) PEG 3350. Well-diffracting crystals of CjLas1-Grc3 complex were grown in a reservoir solution containing 0.1 M sodium phosphate, pH 7.5, 0.05 M NaCl, and 9% (w/v) PEG 4000. The best crystals of CjLas1 HEPN domain protein were grown from 0.1 M Tris–HCl, pH 8.0, 0.2 M MgCl$_2$, and 25% (w/v) PEG 3350. All crystals soaked in cryoprotectants made from the mother liquors supplemented with 20% (v/v) glycerol and flash frozen in liquid nitrogen.

All diffraction datasets were collected at beamline BL-17U1, BL-18U1, and BL-19U1 at the Shanghai Synchrotron Radiation Facility (SSRF) and National Center for Protein Sciences Shanghai (NCPSS), and processed with HKL3000 (*Otwinowski and Minor, 1997*). The CjLas1 HEPN structure was determined by molecular replacement using the HEPN domain structure within CtLas1-Grc3 (PDB: 6OF4) as the search model using the program PHENIX Phaser (*Adams et al., 2002*; *Pillon et al., 2019*). The phases of ScLas1-Grc3 complex and CjLas1-Grc3 complex were solved by molecular replacement method with the cryo-EM maps of Las1-Grc3 complexes using PHENIX Phaser. The model was manually built and adjusted using the program COOT (*Emsley et al., 2010*). Iterative cycles of crystallographic refinement were performed using PHENIX. All data processing and structure refinement statistics are summarized in *Table 1*. Structure figures were prepared using PyMOL (http://www.pymol.org/).

## Cryo-EM data acquisition

The samples were diluted at a final concentration of around 1.0 mg/ml. Next, 3 μl of the samples were applied onto glow-discharged 200-mesh R2/1 Quantifoil copper grids. The grids were blotted for 4 s and rapidly cryocooled in liquid ethane using a Vitrobot Mark IV (Thermo Fisher Scientific) at 4°C and 100% humidity. The samples were imaged in a Titan Krios cryo-electron microscope (Thermo Fisher Scientific) at a magnification of 105,000× (corresponding to a calibrated sampling of 0.82 Å per pixel). Micrographs were recorded using EPU software (Thermo Fisher Scientific) with a K3 detector, where each image was composed of 30 individual frames with an exposure time of 3 s and an exposure rate of 16.7 electrons per second per Å$^2$. A total of 2520 movie stacks for ScLas1-Grc3 complex and 8616 movie stacks for CjLas1-Grc3 complex were collected.

## Single-particle image processing and 3D reconstruction

All micrographs were first imported into Relion (*Scheres, 2012*) for image processing. The motion correction was performed using MotionCor2 (*Zheng et al., 2017*), and the contrast transfer function (CTF) was determined using CTFFIND4 (*Rohou and Grigorieff, 2015*). All particles were autopicked using the NeuralNet option in EMAN2 (*Tang et al., 2007*). Then, particle coordinates were imported to Relion, where the poor 2D class averages were removed by several rounds of 2D classification. Initial maps were built and classified using ab initio 3D reconstruction in cryoSPARC (*Punjani et al., 2017*) without any symmetry applied. Heterogeneous refinement was further performed to remove bad particles using one good and one bad starting map. The good class having 264,341 particles for ScLas1-Grc3 complex or 523,843 particles for CjLas1-Grc3 complex was selected and subjected to 3D homogeneous refinement, local and global CTF refinement, and non-uniform refinement with C2 symmetry imposed, achieving a 3.07 Å resolution map for the ScLas1-Grc3 complex and a 3.39 Å

resolution map for the CjLas1-Grc3 complex, respectively. Resolutions for the final maps were estimated with the 0.143 criterion of the Fourier shell correlation curve. Resolution maps were calculated in cryoSPARC using the 'Local Resolution Estimation' option. (More information in *Figure 2—figure supplement 1*, *Figure 3—figure supplement 1*, and *Table 2*).

### In vitro transcription of RNA

Unlabeled ITS2 RNAs used for cleavage assays were synthesized by in vitro transcription with T7 RNA polymerase and linearized plasmid DNAs as templates. Transcription reactions were performed at 37°C for 4 hr in a buffer containing 100 mM HEPES-KOH, pH 7.9, 20 mM $MgCl_2$, 30 mM DTT, 2 mM each NTP, 2 mM spermidine, 0.1 mg/ml T7 RNA polymerase, and 40 ng/µl linearized plasmid DNA template. The transcribed RNA was then purified by gel electrophoresis on a 12% denaturing (8 M urea) polyacrylamide gel, and RNA band was excised from the gel and recovered with Elutrap System. The purified RNA was resuspended in diethyl pyrocarbonate-treated water.

### In vitro ITS2 RNA cleavage assays

For 5′-Cy5 and 3′-Cy3-labeled ITS2 RNA cleavage assays, 0.5 µM of ITS2 RNA was incubated at 37°C for 2 hr with increasing amounts of Las1, Grc3, or Las1-Grc3 complex proteins (0.05–0.5 µM) in a cleavage buffer containing 20 mM Tris–HCl, pH 7.5, 300 mM NaCl, 1 mM TCEP. 5′-Cy5 and 3′-Cy3-labeled ITS2 RNA was synthesized from Takara Biomedical Technology. Reactions were stopped by adding 2× loading buffer. Samples were analyzed on a 20% urea denaturing polyacrylamide gel with TBE buffer. Cleavage products were visualized by fluorescent imaging and analysis system (SINSAGE Technology).

For unlabeled ITS2 RNA cleavage assays, 10 µM of ITS2 RNA was incubated at 37°C for 2 hr with increasing amounts of Las1, Grc3, or Las1-Grc3 complex proteins (1.5–15 µM) in a cleavage buffer containing 20 mM Tris–HCl, pH 7.5, 300 mM NaCl, 1 mM TCEP. Reactions were stopped by adding 2× loading buffer and were then quenched at 75°C for 5 min. Samples were analyzed on a 20% urea denaturing polyacrylamide gel with TBE buffer. Cleavage products were visualized by toluidine blue staining. The experiment was repeated three times.

### RNA cleavage product phosphorylation assays

A 10 µM unlabeled ITS2 RNA was incubated at 37°C for 1.5 hr with 10 µM of Las1-Grc3 complex proteins in a cleavage buffer containing 20 mM Tris–HCl, pH 7.5, 300 mM NaCl, 1 mM TCEP. Then, 1 mM ATP was added into each reaction system at 37°C for 30 min. Reactions were stopped by adding 2× loading buffer and were then quenched at 75°C for 5 min. Samples were analyzed on a 20% urea denaturing polyacrylamide gel with TBE buffer. Cleavage products were visualized by toluidine blue staining. The experiment was repeated three times.

### GST pull-down assays

Pull-down experiments were carried out using GST fusion proteins to analyze the association between Las1 and Grc3. Then, 0.2 mg purified GST-ScLas1 protein was incubated with 0.4 mg purified ScGrc3 protein in binding buffer containing 20 mM Tris–HCl, pH 7.5, 500 mM NaCl, 2 mM DTT. Also, 80 µl GST affinity resin was added into each reaction system at 4°C for 60 min. The resin was washed with 1 ml of binding buffer. After washing five times, the binding samples were eluted with elution buffer containing 20 mM Tris, pH 7.5, 20 mM GSH, 500 mM NaCl, and 2 mM DTT. The elution samples were then monitored using SDS-PAGE and visualized by Coomassie blue staining. The assays were quantified by band densitometry. The experiment was repeated three times.

### Electrophoretic mobility shift assays

EMSA were performed with a series of Las1-Grc3 complex dilutions from 20 µM to 2 µM and an 81-nt ITS2 RNA. The ITS2 RNA was synthesized by in vitro transcription. Proteins were incubated with ITS2 RNA in a binding buffer containing 20 mM Tris–HCl, pH 7.5, 300 mM NaCl for 30 min at 4°C. After the reaction, the binding samples were then resolved on 5% native acrylamide gels in Tris-Glycine (0.5× TBE) buffer pH 8.5 under an electric field of 100 V for 40 min at 4°C. Gels were imaged by using a ChemiDoc XRS+ (Bio-Rad). The experiment was repeated three times.

## RNA sequencing

Enzyme-digested products from the ITS2 cleavage assay were isolated by dialysis from denaturing polyacrylamide gel. T4 PNK was used to phosphorylate the 5′ terminus of RNA products. After the above treatment, RNA samples were linked with the adaptor RNA by T4 RNA ligase-1. The RNA samples were then treated for reverse transcription and PCR amplification. The PCR amplification products were cloned into pET28a vector, then sequenced. The experiment was repeated three times.

## Acknowledgements

We are grateful to the staff of the BL-17U1, BL-18U1, and BL-19U1 beamlines at the National Center for Protein Sciences Shanghai (NCPSS) at Shanghai Synchrotron Radiation Facility (SSRF). We thank the Cryo-EM Center at the University of Science and Technology of China for the support of cryo-EM data collection.

## Additional information

### Funding

| Funder | Grant reference number | Author |
| --- | --- | --- |
| National Natural Science Foundation of China | 32171286 | Liang Liu |
| National Natural Science Foundation of China | 32022047 | Liang Liu |
| The Ministry of Science and Technology of China | 2022YFC2303700 | Kaiming Zhang Shanshan Li |
| The Ministry of Science and Technology of China | 2022YFA1302700 | Kaiming Zhang |
| The Strategic Priority Research Program of the Chinese Academy of Sciences | XDB0490000 | Kaiming Zhang |
| Center for Advanced Interdisciplinary Science and Biomedicine of IHM | QYPY20220019 | Kaiming Zhang |
| The Fundamental Research Funds for the Central Universities | WK9100000044 | Kaiming Zhang |
| The Fundamental Research Funds for the Central Universities | WK9100000032 | Shanshan Li |

The funders had no role in study design, data collection and interpretation, or the decision to submit the work for publication.

### Author contributions

Jiyun Chen, Validation, Investigation, Methodology, Project administration; Hong Chen, Shanshan Li, Xiaofeng Lin, Validation, Investigation; Rong Hu, Investigation; Kaiming Zhang, Supervision, Funding acquisition, Investigation, Methodology, Writing - original draft, Writing - review and editing; Liang Liu, Supervision, Funding acquisition, Validation, Investigation, Methodology, Writing - original draft, Project administration, Writing - review and editing

### Author ORCIDs

Hong Chen http://orcid.org/0000-0002-0179-4493
Liang Liu http://orcid.org/0000-0002-5379-0638

Reviewer #1 (Public Review): https://doi.org/10.7554/eLife.86847.3.sa1

Reviewer #2 (Public Review): https://doi.org/10.7554/eLife.86847.3.sa2
Author Response https://doi.org/10.7554/eLife.86847.3.sa3

## Additional files

### Supplementary files
• MDAR checklist

### Data availability

The atomic coordinates of the reported X-ray and cryo-EM structures have been deposited in the Protein Data Bank (PDB) with the following accession codes: 7Y16 (CjLas1, X-ray), 7Y17 (CjLas1-Grc3 complex, X-ray), 7Y18 (ScLas1-Grc3 complex, X-ray), 8J5Y (ScLas1-Grc3 complex, cryo-EM), and 8J60 (CjLas1-Grc3 complex, cryo-EM). Cryo-EM maps of the ScLas1-Grc3 complex and CjLas1-Grc3 complex in this study have been deposited in the wwPDB OneDep System under EMD accession codes EMD-33733 and EMD-33735.

The following datasets were generated:

| Author(s) | Year | Dataset title | Dataset URL | Database and Identifier |
|---|---|---|---|---|
| Chen J, Liu L | 2023 | Crystal structure of rRNA-processing protein Las1 | https://www.rcsb.org/structure/7Y16 | RCSB Protein Data Bank, 7Y16 |
| Chen J, Liu L | 2023 | Crystal structure of ribosomal ITS2 pre-rRNA processing complex from Cyberlindnera jadinii | https://www.rcsb.org/structure/7Y17 | RCSB Protein Data Bank, 7Y17 |
| Chen J, Liu L | 2023 | Crystal structure of ribosomal ITS2 pre-rRNA processing complex from Saccharomyces cerevisiae | https://www.rcsb.org/structure/7Y18 | RCSB Protein Data Bank, 7Y18 |
| Chen J, Chen H, Li S, Lin X, Hu R, Zhang K, Liu L | 2023 | Structural and mechanistic insight into ribosomal ITS2 RNA processing by nuclease-kinase machinery | https://www.rcsb.org/structure/8J5Y | RCSB Protein Data Bank, 8J5Y |
| Chen J, Chen H, Li S, Lin X, Hu R, Zhang K, Liu L | 2023 | Structural and mechanistic insight into ribosomal ITS2 RNA processing by nuclease-kinase machinery | https://www.rcsb.org/structure/8J60 | RCSB Protein Data Bank, 8J60 |
| Chen J, Chen H, Li S, Lin X, Hu R, Zhang K, Liu L | 2023 | Cryo-EM structure of ScLas1-Grc3 complex | https://www.ebi.ac.uk/emdb/EMD-33733 | Electron Microscopy Data Bank, EMD-33733 |
| Chen J, Chen H, Li S, Lin X, Hu R, Zhang K, Liu L | 2023 | Cryo-EM structure of CjLas1-Grc3 complex | https://www.ebi.ac.uk/emdb/EMD-33735 | Electon Microscopy Data Bank, EMD-33735 |

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
