## [Editor Report · eLife assessment]

This study represents a **valuable** mechanistic contribution towards understanding how ribosomal RNA is processed during ribosome biogenesis. The biochemical evidence supporting the major conclusions is **convincing**. This work will be of interest to cell biologists and biochemists working on ribosome biogenesis.

---

## [Referee Report · Reviewer #1 (Public Review)]

The findings in this paper can be split into three parts.

1. Processing of ITS2

Firstly, the authors identify two sites on ITS2 which are cleaved by the ScLas1-Grc3 complex, as part of 25S ribosomal RNA maturation.

For a smaller segment of ITS2 (33nt), the two sites separate out 3 parts with sizes of 10nt, 14nt, and 9nt (Figure 1C). However, bands in mass spec. occur at 23nt (14nt+9nt) and 14nt, but not 9nt alone. Additional bands can be seen at 22nt and 21nt. Hence the evidence for these two specific sites seems somewhat uncertain. It is not clear if there is an experimental limitation in terms of accuracy, or that the cleavage is perhaps somewhat approximate at the two sites. The authors may try to clarify these results a bit further.

For a larger ITS2 (81nt), similar support is found for the two cleavage sites, but now the possible fragments are 14nt, 30nt, 37nt, and 44nt (14nt+30nt) (Figure 1E). The observed bands match these fragments at 44nt, 37nt, 30nt, but again there are additional bands at 36nt, 28nt, and 13nt, which are not fully explained. It may be useful for explain or discuss these discrepancies.

Another useful result of these experiments is to confirm that Las1 alone has only weak activity against ITS2, but very strong activity when it is part of the Las1-Grc3 complex.

1. Structure of Las1, and Las1-Grc3 complexes

A second important contribution of this work are X-ray and cryoEM structures of Las1-Grc3 from Sc and Cj. It is interesting that even though the complexes are very similar, CjGrc3 shows weak activation of CjLas1, whereas ScGrc3 more strongly activates ScLas1. The X-ray and cryoEM structures are very similar. However, the X-ray structures also show an additional (CC) domain from Las1 not resolved in the cryoEM map. This difference is significant, because it suggests the CC domain may remain more flexible in solution, but stabilizes in the crystal. Also interestingly, the CC domains have different structures and are in different positions in ScLas1-Grc3 vs CjLas1-Grc3, again hinting that they are more dynamic. Further experiments described by the authors confirm the CC domain is indeed important in RNA binding and activity. Whether they are only implicated in binding RNA or both binding and cleavage is somewhat unclear.

The structure of Las1-Grc3 is described as resembling a butterfly, with Las1 being the body and Grc3 the wings. While this is a useful description, it may be a bit misleading. The complex has C2 symmetry, with one Las1-Grc3 unit related to the other by about ~180 rotation around a vertical axis parallel to the body of the butterfly as proposed. To use the butterfly analogy, one half of the body and one wing faces the opposite way as the other, not a mirror symmetry as a real butterfly would have.

Both Cj and Sc structures show the C-terminal of Grc3 binding to the active pocket of Las1, explaining its effect on activity. Mutation experiments also further show the importance of these residues on activity. Reciprocally, a region in Las1, LCT, inserts into Grc3, forming a stable complex. Again mutating these residues affects activity, strengthening their importance and the evidence for how the stable complex forms.

Finally, an X-ray structure of dimeric Las1 in Cj, without Grc3 is presented. Truncating CC and LCT appeared to be necessary to allow the dimer to crystallize. Superposition with Las1 dimer in Las1-Grc3 shows a conformational difference, and different distances between residues in the active pocket, explaining the change in activity with and without Grc3. Interestingly, the Las1 domains themselves do not change too much, i.e. both domains can be matched with less than 1Å Ca-RMSD, so the difference may be more of a repositioning of the two domains for the active conformation.

One notable strength of this study is the use of both X-ray and cryoEM to obtain structures of the Las1-Grc3 and dimeric Las1 complexes. Typically structures of cryoEM at ~3Å are sufficient for reliable modeling; for example, the backbone and side chains of residues in the active site are well resolved. However, in this case, a cryoEM model of the Las1 dimer was not obtained, so it was important to show first that the Las1-Las1 conformation in the Las1-Grc3 complex is the same in both X-ray and cryoEM models. Otherwise, there may remain doubt whether the X-ray model of Las1 dimer could be compared to the cryoEM map of Las1-Grc3, as crystallization conditions could potentially influence conformation and arrangement. It would be interesting to know whether a cryoEM structure of the Las1 dimer alone was attempted - perhaps it was too small to be reliably seen in micrographs. Having had such a model could avoid the need of X-ray structures, although of course more experimental data are always useful.

1. Mechanism of ITS2 cleavage

The proposed mechanism shown in Figure 8 seems to be well supported by the obtained structures and biochemical experiments. A question that remains is why it is proposed that both C2 and C2' cleavage can be performed upon a single binding of the ITS2 RNA, i.e. seeming to suggest there are two binding sites. This would seem to directly generate 3 fragments, without any other intermediate products. Mass spec. seemed to show the intermediate products, perhaps indicating two binding events for each cleavage process. Perhaps the authors could discuss this more. Also perhaps can be good to discuss whether it would be possible to obtain a structure with the bound RNA, further giving structural information of how the exact cleavage process is performed.

---

## [Referee Report · Reviewer #2 (Public Review)]

In this manuscript, Chen et al. determined the structural basis for pre-RNA processing by Las1-Grc3 endoribonuclease and polynucleotide kinase complexes from *S. cerevisiae* (Sc) and C. jadinii (Cj). Using a robust set of biochemical assays, the authors identify that the sc- and CjLas1-Grc3 complexes can cleave the ITS2 sequence in two specific locations, including a novel C2' location. The authors then determined X-ray crystallography and cryo-EM structures of the ScLas1-Grc3 and CjLas1-Grc3 complexes, providing structural insight that is complimentary to previously reported Las1-Grc3 structures from C. thermophilum (Pillon et al., 2019, NSMB). The authors further explore the importance of multiple Las1 and Grc3 domains and interaction interfaces for RNA binding, RNA cleavage activity, and Las1-Grc3 complex formation. Finally, evidence is presented that indicates Las1 undergoes a conformational change upon Grc3 binding that stabilizes the Las1 HEPN active site, providing a possible rationale for the stimulation of Las1 cleavage by Grc3.

In the revised manuscript, the authors have made significant efforts towards addressing initial reviewer comments. This includes further clarification for key biochemical experiments, significant improvement in structural model quality, and additional structural analysis that further strengthens major conclusions in the manuscript. Overall, the authors conclusions are now well supported by the biochemical and structural data provided.

---

## [Author Response]

The following is the authors’ response to the original reviews.

**Public Reviews:**

**Reviewer #1 (Public Review):**
In the manuscript there is not much comparison between the crystal and cryoEM structures provided, and on inspection they appear to be very similar. The crystal structures also reveal parts of the CC domains in Las1, which is not present in the cryoEM structures. It is interesting the CC domains in Sc and Cj are quite different as illustrated in Figure 4B. They also seem to be somewhat disconnected from the rest of the complex (more so for Cj), even though that's not apparent in Figures 2-4. Despite this, it would be very useful to show the cryoEM densities when describing the catalytic site and C-terminal domain interactions, for example, as this can be very useful to increase confidence in the model and proposed mechanisms.

We thank the reviewer for this suggestion. We have added a figure (Figure 5- Figure supplement 3) to show cryo-EM and crystal densities of key amino acids, when describing the catalytic site and C-terminal domain interactions. In analyzing the interaction between Las1 and Grc3, we have also provided additional comparisons of the crystal structure and the cryo-EM structure (Figure 5, Figure 5-figure supplement 1, 2 and 3, Figure 6, Figure 6-figure supplement 1).

The description of the complex as a butterfly is engaging, and from a certain angle it can be made to look as such; this was also described previously in (Pillon et al., 2019, NSMB) for the same complex from a different organism (Ct). However, it is a bit misleading, because the complex is actually C2 symmetric. Under this symmetry, the 'body' would consist of two 'heads' one pointing up, one down facing towards the back, and one wing would have its back toward the viewer, the other the front. The structures presented here in Sc and Cj seem quite similar to the previous structure of the same complex in Ct, though the latter was only solved with cryoEM, and was also lacking the structure of the CC domain in Las1.

We thank the reviewer for pointing out this issue. We have re-wrote these sentences and changed the butterfly description of Las1-Grc3 complex in the revised manuscript.

For the model suggested in Figure 8, perhaps in the 'weak activity' state, the LCT in Las1 could still be connected to Grc3, via the LCT, rather than disconnected as shown. This could facilitate faster assembly of the 'high activity' state. The complex is described as 'compact and stable', but from the structure and this image, it appears more dynamic, which would serve its purpose and the illustrated model better. The two copies of HEPN appear to have more connective area, meaning they are indeed more likely to remain assembled in the 'weak activity' state. On the other hand, HEPN in one protein appears to have less binding surface with PNK in Grc3, and even less so with the CTD (both PNK and CTD being from the other associated protein), meaning these bindings could release easily to form the 'weak activity' state.There is also the potential to speculate that the GCT is bound to HEPN near the catalytic area in the 'weak activity' state. The reduced activity when the GCT residues are replaced by Alanine could then be explained by the complex not being able to assemble as quickly upon binding of the substrate, as it could if the GCT remained bound, rather than by a conformational change that it induces upon binding. The conformational change is also likely to be influenced by the combined binding of PNK and CTD in the assembled state, which also contact HEPN, rather than by GCT alone.

We thank the reviewer for this suggestion. We have revised our model in the new Figure 8 of our revised manuscript. We apologize for the un-clarity description of the 'weak activity' state in our model. In fact, we believe that Las1 is in a "weakly activity" state before binding to Grc3 and is in a "highly activity" state when it forms a complex with Grc3. We strongly agree that the Las1-Grc3 complex is more dynamic than compact and stable, so it is easy to change its active state. We have changed our description and revised our model in the revised manuscript.

When comparing the structure of the HEPN domain in the lone Las1 protein to the structure of Las1-HEPN in the Las1-Grc3 complex, it is mentioned that 'large conformational changes are observed'. These could be described a bit better. The conformational change is ~3-4Å C-alpha RMSD across all ~150 residues in the domain (~90 residues forming a stable core that only changes by ~1Å). There is also a shift in the associated HEPN domain in Las1B domain compared to the bound HEPN in the Las1-Grc3 complex, as shown in Figure 7D: ~1Å shift and ~12degrees rotation. This does point to the conformation of HEPN changing upon complex formation, as does the relative positions of the HEPN domains in Las1A and Las1B. The conformational change and relative shift could indeed by key for the catalysis of the substrate as mentioned.

We thank the reviewer for this great suggestion. We have replaced the sentence describing the conformational changes in our revised manuscript.

Overall, the structures presented should be very useful in further study of this system, even though the exact dynamics and how the substrate is bound are aspects that are perhaps not fully clear yet. The addition of the structures of the CC domain in two different organisms and the Las1 HEPN domain (not in complex with Grc3) as new structural information should allow for increasing our understanding of the overall complex and its mechanism.

We thank this reviewer for these encouraging comments, which helped us with greatly improving our manuscript.

**Reviewer #2 (Public Review):**
In this manuscript, Chen et al. determined the structural basis for pre-RNA processing by Las1-Grc3 endoribonuclease and polynucleotide kinase complexes from *S. cerevisiae* (Sc) and C. jadinii (Cj). Using a robust set of biochemical assays, the authors identify that the sc- and CjLas1-Grc3 complexes can cleave the ITS2 sequence in two specific locations, including a novel C2' location. The authors then determined X-ray crystallography and cryo-EM structures of the ScLas1-Grc3 and CjLas1-Grc3 complexes, providing structural insight that is complimentary to previously reported Las1-Grc3 structures from C. thermophilum (Pillon et al., 2019, NSMB). The authors further explore the importance of multiple Las1 and Grc3 domains and interaction interfaces for RNA binding, RNA cleavage activity, and Las1-Grc3 complex formation. Finally, evidence is presented that suggests Las1 undergoes a conformational change upon Grc3 binding that stabilizes the Las1 HEPN active site, providing a possible rationale for the stimulation of Las1 cleavage by Grc3.Several of the conclusions in this manuscript are supported by the data provided, particularly the identification and validation of the second cleavage site in the ITS2. However, several aspects of the structural analysis and complimentary biochemical assays would need to be addressed to fully support the conclusions drawn by the authors.

We thank the reviewer for the positive comments.

• There is a lack of clarity regarding the number of replicates performed for the biochemical experiments throughout the manuscript. This information is critical for establishing the rigor of these biochemical experiments.

We apologize for not providing the detailed information on the number of replicates of biochemical experiments. All the biochemical experiments were repeated three times. We have provided this information in the figure legends.

• The authors conclude that Rat1-Rai1 can degrade the phosphorylated P1 and P2 products of ITS2 (lines 160-162, Figure 1H). However, the data in Fig. 1H shows complete degradation of 5'Phos-P2 and 5'Phos-P4 of ITS2, while the P1 and 5'Phos-P3 fragments remain in-tact. Additional clarification for this discrepancy should be provided.

We thank the reviewer for pointing out this issue. “phosphorylated P1 and P2 products” should be “phosphorylated P2 and P4 products”. We have corrected this clerical error. In addition, we have also provided an explanation for why phosphorylated P3 product show only partial degradation. We suspect that P3 product may be too short to completely degrade.

• The authors determined X-ray crystal structures of the ScLas1-Grc3 (PDB:7Y18) and CjLas1-Grc3 (PDB:7Y17) complexes, which represents the bulk of the manuscript. However, there are major concerns with the structural models for ScLas1-Grc3 (PDB:7Y18) and CjLas1-Grc3 (PDB:7Y17). These structures have extremely high clashscores (>100) as well as a significant number of RSRZ outliers, sidechain rotamer outliers, bond angle outliers, and bond length outliers. Moreover, both structures have extensive regions that have been modeled without corresponding electron density, and other regions where the model clearly does not fit the experimental density. These concerns make it difficult to determine whether the structural data fully support several of the conclusions in the manuscript. A more careful and thorough reevaluation of the models is important for providing confidence in these structural conclusions.

We thank the reviewer for pointing out this issue. We have used the cryo-EM datasets to further validate our conclusions of the manuscript. We analyzed the active site of Las1-Grc3 complex and the interactions between Las1 and Grc3 using the cyro-EM structures and presented new figures (Figure 5- Figure supplement 1, Figure 5- Figure supplement 2, Figure 5- Figure supplement 3, Figure 6- Figure supplement 1) in our revised manuscript. Both the refinement and validation statistical parameters of the cryo-EM datasets are within a reasonable range (Table 2), which will provide confidence for our structure conclusions. The X-ray crystal structures of ScLas1-Grc3 (PDB:7Y18) and CjLas1-Grc3 (PDB:7Y17) complexes has high calshscores and many outliers, which is mainly due to the great flexibility of Las1-Grc3 complex, especially the CC domain of Las1. We have improved our crystal structure models with better refinement and validation of statistical parameters. The clashscores of ScLas1-Grc3 complex and CjLas1-Grc3 complex are 25 and 45, respectively. There are no rotamer outliers and C-beta outliers to report for both ScLas1-Grc3 complex and CjLas1-Grc3 complex.

• The presentation of the cryo-EM datasets is underdeveloped in the results section drawing and the contribution of these structures towards supporting the main conclusions of the manuscript are unclear. An in-depth comparison of the structures generated from X-ray crystallography and cryo-EM would have greatly strengthened the structural conclusions made for the ScLas1-Grc3 and CjLas1-Grc3 complexes.

We thank the reviewer for this suggestion. We have performed structural comparisons between X-ray crystal structure and cyro-EM structure in analyzing the active site of Las1-Grc3 complex and the interactions between Las1 and Grc3 (Figure 5- Figure supplement 1, Figure 5- Figure supplement 2, Figure 6- Figure supplement 1). We have also added a figure (Figure 5- Figure supplement 3) to show cryo-EM and crystal densities of the Las1 active site as well as the key amino acids for Las1 and Grc3 interactions. These comparisons and densities have greatly strengthened our structural conclusions.

• The authors conclude that truncation of the CC-domain contributes to Las1 IRS2 binding and cleavage (lines 220-222, Fig. 4C). However, these assays show that internal deletion of the CC-domain alone has minimal effect on cleavage (Fig 4C, sample 3). The loss in ITS2 cleavage activity is only seen when truncating the LCT and LCT+CC-domain (Fig 4C, sample 2 and 4, respectively). Consistently, the authors later show that Las1 is unable to interact with Grc3 when the LCT domain is deleted (Fig. 6 and Fig. 6-figure supplement 2). These data indicate the LCT plays a critical role in Las1-Grc3 complex formation and subsequent Las1 cleavage activity. However, it is unclear how this data supports the stated conclusion that the CC-domain is important for LasI cleavage.

Our EMSA data shows that the CC domain contributes to the binding of ITS2 RNA (Figure 4D), suggesting that the CC domain may play a role of ITS2 RNA stabilization in the Las1 cutting reaction. The in vitro RNA cleavage assays (Figure 4C) indicate that the LCT is important for Las1 cleavage because it plays a critical role in the formation of the Las1-Grc3 complex. Compared with LCT, the CC domain, although not particularly important for Las1 cutting ITS2, still has some influence (Fig 4C, sample 1 and 3, sample 2 and 4,). Therefore, we conclude that the CC domain may mainly play a role in the stabilization of ITS2 RNA, thereby enhancing ITS2 RNA cleavage.

• The authors conclude that the HEPN domains undergo a conformational change upon Grc3 binding, which is important for stabilization of the Las1 active site and Grc3-mediated activation of Las1. This conclusion is based on structural comparison of the HEPN domains from the CjLas1-Grc3 complex (PDB:7Y17) and the structure of the isolated HEPN domain dimer (PDB:7Y16). However, it is also possible that the conformational changes observed in the HEPN domain are due to truncation of the Las1 CC and CGT domains. A rationale for excluding this possibility would have strengthened this section of the manuscript.

We thank the reviewer for pointing out this issue. We agree that the complete Las1 structure information is helpful in illuminating the conformational activation of the Las1 by Grc3. We screened about 1200 crystallization conditions with full-length Las1 proteins, but ultimately did not obtain any crystals, probably due to flexibility. The CC domain exhibits a certain degree of flexibility, which has not been observed in the structure obtained from electron microscopy. The LCT is involved in binding to the CTD domain of Grc3. The coordination of the active center of HEPN domains by LCT and CC domains is unlikely due to the limited nuclease activity observed in full-length Las1. The conformational changes of the active center are essential for HEPN nuclease activation. Our structure shows that the GCTs of Grc3 interact with the active residues of Las1 HEPN domains, which probably induce conformational changes in the active center of the HEPN domain to activate Las1. Of course, we cannot exclude the possibility that truncation of the Las1 CC and LCT domains will result in little conformational change in the HEPN domains. We have explained this possibility in our revised manuscript.

**Reviewer #1 (Recommendations For The Authors):**
1. It would be very useful to show the cryoEM densities when describing the catalytic site and C-terminal domain interactions.

The new Figure 5-figure supplement 2 have showed the Cyro-EM densities of the catalytic site of ScLas1 and the C-terminal domain of ScGrc3.

1. "ScLas1 cleaves the 33-nt ITS2 at C2 site to theoretically generate a 10-nt 5′-terminal product and a 23-nt 3′-terminal product (Figure 1A). Our merger data shows that the final 5′-terminal and 3′-terminal product bands are at nearly the same horizontal position on the gel (Figure 1B), indicating that they are similar in size."These two sentences seem to contradict, i.e. 10-nt and 23-nt are similar in size even though they are different lengths?

We apologize for the contradiction in these two sentences mentioned above. We have re-wrote these two sentences in the revised manuscript.

1. We observed four cleavage bands of approximately 23-nt (P2), 14-nt (P3), 10-nt (P1), and 9-nt (P4) in length (Figure 1C). "Figure 1C. The bands show 23 nt, 22 nt, 21nt, 14 nt, 13nt, and 11nt, so this text does not seem to describe the figure.

We have re-wrote this sentence in the revised manuscript.

1. "We obtained similar cleavage results with a longer 81-nt ITS2 RNA substrate 6 (Figure 1D, E). "Figure 1D,E. The lengths in Figure 1E do not correspond to all bands in Figure 1E, e.g. the 13 nt band, though the others do, e.g. 14 nt, 30nt, 37nt, etc.

In order to better evaluate the size of the cut product, we used an RNA marker as a comparison. The RNA marker will have more bands than the cleavage products. To further confirm the cleavage site of C2′, we also mapped the cleavage sites of the 81-nt ITS2 using reverse transcription coupling sequencing methods (Figure 1F).

1. In Figure 3, domains are colored different but it's hard to know which are different proteins.

We have added a diagram in Figure 3 to show the Las1-Grc3 complex structure, and it is now clear how Las1 and Grc3 are assembled into a tetramer.

1. Line 267. "we screened a lot of crystallization conditions with full-length Las1 proteins"How many? Rough numbers ok, but 'a lot' is not very informative

We have provided the approximate numbers of crystallization conditions in our revised manuscript.

**Reviewer #2 (Recommendations For The Authors):**
1. The authors missed an excellent opportunity to compare and contrast the ScLas1-Grc3 and CjLas1-Grc3 complex structures presented here with that of the previously determined CtLas1-Grc3 structure (Pillon et al., 2019, NSMB). For example, His130 in the ScLas1-Grc3 complex active site adopts a similar conformation to His142 in the TcLas1-Grc3 complex active site (Pillon et al., 2019, NSMB). Interestingly, the analogous His134 active site residue in the CjLas1-Grc3 adopts an alternative (maybe inactive) conformation. This observation could provide a structural rationale for the activation of scLas1 and TcLas1 by Grc3, while also providing a rationale for the fairly weak activation of CjGrc3 by CjGrc3.

We thank the reviewer for this suggestion. We have performed structural comparisons between ScLas1-Grc3, CjLas1-Grc3 and CtLas1-Grc3 complexes, especially the Las1 nuclease active center. We added two figures (Figure7-figure supplement 3A and 3B) in the revised manuscript to contrast and highlight the conformational differences of active amino acids in active centers between ScLas1-Grc3, CtLas1-Grc3 and CjLas1-Grc3. These structural comparisons provide stronger evidence that further reinforces the conclusions of our manuscript.

1. Can the authors speculate as to whether the structural data can provide any insight into how the Las1-Grc3 may cleave both C2 and C2' positions in the ITS2 RNA? This commentary would further strengthen the discussion section of the manuscript.

We thank the reviewer for this suggestion. We have provided a speculation in the discussion section of the revised manuscript.

We think that the structural data may provide some insight into how Las1-Grc3 complex cleaves ITS2 RNA at both C2 and C2' positions. The Las1-Grc3 tetramer complex has one nuclease active center and two kinase active centers. The nuclease active center consists of two Las1 molecules in a symmetric manner, while the kinase active center consists of only one Grc3 molecule. The ITS2 RNA is predicted to form a stem-loop structure. The symmetrical nuclease active center recognizes the stem region of ITS2 RNA and makes it easy to perform C2 and C2' cleavages on both sides of the stem. C2 and C2' cleavage products are further phosphorylated by two Grc3 kinase active centers, respectively.

1. The method used for the plasmid generation, expression, and purification of the Las1 truncations and the Las1 and Grc3 point mutants should be provided in the methods section.

The method used for the plasmid generation, expression, and purification of the Las1 truncations and the Las1 and Grc3 point mutants have be provided in the methods section.

1. The exact amino acid cutoffs for the truncated forms of Las1 used for the biochemical assays in Fig. 4 should be provided.

We have provided the exact amino acid cutoffs for the truncated forms of Las1 in the figure legend of Figure 4C.

1. The models associated with the cryo-EM datasets should be deposited in the PDB.

The models associated with the Cryo-EM datasets have be deposited in the PDB with the following accession codes: 8J5Y (ScLas1-Grc3 complex), and 8J60 (CjLas1-Grc3 complex).

1. Lines 232-234: Arg129 should be changed to His134.

We have corrected it.

1. Figure 5B: the bottom half of the HEPN active site has been labeled incorrectly. The labels should be Arg129, His130, and His134 (from left to right).

We have corrected it.

1. Line 252: "multitudinous" should be changed to "multiple."

We have corrected it.